# Balancing central control and sensory feedback produces adaptable and robust locomotor patterns in a spiking, neuromechanical model of the salamander spinal cord

**Alessandro Pazzaglia** [iD][1]◉*, **Andrej Bicanski** [iD][2]◉, **Andrea Ferrario** [iD][1], **Jonathan Arreguit**[1], **Dimitri Ryczko**[3], **Auke Ijspeert**[1]

**1** Biorobotics Laboratory, École Polytechnique Fédérale de Lausanne (EPFL), Lausanne, Switzerland, **2** Neural Computation Group, Max Planck Institute for Human Cognitive and Brain Sciences, Leipzig, Germany, **3** Ryczko Laboratory, Department of Pharmacology-Physiology, Université de Sherbrooke, Sherbrooke, Canada

◉ These authors contributed equally to this work.
* alessandro.pazzaglia@epfl.ch

**Data availability statement:** All data and code used for running simulations, analyses, and plotting is available on a GitHub repository at https://github.com/AlexPazzaglia/SalamandraSNN.

**Funding:** The work of AP, AF, JA, DR and AI was supported by the European Research Council (ERC) under the European Union's Horizon 2020 research and innovation

## Abstract

This study introduces a novel neuromechanical model employing a detailed spiking neural network to explore the role of axial proprioceptive sensory feedback, namely stretch feedback, in salamander locomotion. Unlike previous studies that often oversimplified the dynamics of the locomotor networks, our model includes detailed simulations of the classes of neurons that are considered responsible for generating movement patterns. The locomotor circuits, modeled as a spiking neural network of adaptive leaky integrate-and-fire neurons, are coupled to a three-dimensional mechanical model of a salamander with realistic physical parameters and simulated muscles. In open-loop simulations (i.e., without sensory feedback), the model replicates locomotor patterns observed in-vitro and in-vivo for swimming and trotting gaits. Additionally, a modular descending reticulospinal drive to the central pattern generation network allows to accurately control the activation, frequency and phase relationship of the different sections of the limb and axial circuits. In closed-loop swimming simulations (i.e. including axial stretch feedback), systematic evaluations reveal that intermediate values of feedback strength increase the tail beat frequency and reduce the intersegmental phase lag, contributing to a more coordinated, faster and energy-efficient locomotion. Interestingly, the result is conserved across different feedback topologies (ascending or descending, excitatory or inhibitory), suggesting that it may be an inherent property of axial proprioception. Moreover, intermediate feedback strengths expand the stability region of the network, enhancing its tolerance to a wider range of descending drives, internal parameters' modifications and noise levels. Conversely, high values of feedback strength lead to a loss of controllability of the network and a degradation of its locomotor performance. Overall, this study highlights the beneficial role of proprioception in generating, modulating and stabilizing locomotion

programme (SALAMANDRA project, grant agreement No. 951477 to DR and AI, https://cordis.europa.eu/project/id/951477). The work of AB was supported by the Max Planck Society (https://www.mpg.de/en).The funders had no role in study design, data collection and interpretation, or the decision to submit the work for publication.

**Competing interests:** The authors have declared that no competing interests exist.

patterns, provided that it does not excessively override centrally-generated locomotor rhythms. This work also underscores the critical role of detailed, biologically-realistic neural networks to improve our understanding of vertebrate locomotion.

## Author summary

In this paper, we developed a computational model to investigate how salamanders move, both while swimming and walking. Unlike previous studies that often oversimplified the dynamics of these complex neural networks, our model includes detailed simulations of the classes of neurons that are considered responsible for generating movement patterns. The locomotor circuits, modeled as a spiking neural network, are coupled to a three-dimensional mechanical model of a salamander with realistic physical parameters and simulated muscles. The neural model integrates axial proprioceptive sensory feedback from the body's movements to modulate the locomotor gaits. Our simulations suggest that this sensory feedback plays a major role in controlling the rhythm and coordination of movements. This has implications for understanding not only how salamanders move but also provides insights into the evolution of locomotion in vertebrates. By investigating how central and sensory mechanisms interact to produce efficient and adaptable movement, our work contributes to the broader field of neuroscience and robotics, offering potential strategies for designing more effective biomimetic robots.

## Introduction

Animal locomotion depends on the coordinated interplay between body mechanics and the neural circuits in the brain, spinal cord, and sensory systems. Decades of experiments and modeling studies have established that specialised spinal circuits called central pattern generators (CPGs) are capable of generating rhythmic outputs underlying motor skills, such as walking and swimming, in the absence of rhythmic inputs (for a review, see [1]). The numerous types of CPG circuits and their respective locomotion patterns were described across multiple vertebrate species, such as swimming in the lamprey [2–5], zebrafish [6–8] and tadpole [9,10], walking and swimming in the salamander [11,12], as well as walking/running in the mouse [13–15] and cat [16,17].

The basic repertoire of motor outputs and several anatomical and genetic properties of CPG, brain stem and sensory circuits appear to be largely conserved across vertebrate species [1,12,18–22]. However, evolution also gave rise to specialized CPG and sensorimotor circuits that allowed new species to adapt to their habitat and to meet the specific needs of each species. A general view is that CPG circuits have been genetically hard coded during evolution to generate the central "stereotypical" motor programs which allow actions to be executed, while sensory feedback loops fine tune these programs and contribute to select new motor programs [23]. However, several studies suggest that sensory feedback can have a bigger role than expected, by contributing to the generation of rhythmic motor outputs in different spinal body parts and to their coordination, especially in large and slow animals [24–28]. Thus it has become increasingly clear that understanding animal locomotion requires studying the interaction between the central and sensory feedback systems in closed loop.

Recent advances in molecular genomics and optogenetics - mostly in the mouse and the zebrafish - combined with traditional electrophysiological and anatomical experiments have identified anatomically and functionally distinct reticulospinal and spinal CPG neuronal populations, which are shared across these and other vertebrate species [5,22,29]. Reticulospinal neurons (RS) in the brain stem relay motor commands from the mesencephalic locomotor region (MLR) to the spinal interneurons to trigger/terminate a locomotion episode, control the steering, frequency and speed of locomotion and change the expressed locomotor pattern [30–36]. Similarly, in the salamander, low level stimulation in the MLR evokes slow walking, while high level stimulation evokes swimming at higher movement frequencies [37]. Genetically identified classes of CPG interneurons are responsible for the coordination of different body parts during locomotion [22]. For example, in the zebrafish and tadpoles, chains of commissural glycinergic V0d neurons and ipsilateral glutamatergic CHx10 positive V2a (called dINs in tadpoles) neurons are responsible for the generation of the alternating left-right rhythm and forward wave propagation during swimming. More complex circuits consisting of multiple cell types are responsible for the coordination between limb CPGs during quadrupedal gaits in the mouse [15,38–40].

In addition to brainstem and CPG circuits, spinal cord circuits are equipped with sensory feedback loops originating from muscles, cutaneous afferents and intraspinal sensory neurons that modify the CPG locomotor patterns according to the interaction with the external world [23]. Proprioceptive (curvature) sensory neurons (PS) in the lamprey and the zebrafish detect stretch and stretch rate within the internal region of the spinal cord [41–44]). Interestingly, while PS neurons in the lamprey (also called edge cells) are made up of distinct excitatory ipsilateral and inhibitory commissural populations [41,45], PS neurons in the zebrafish are exclusively commissural and inhibitory and connect preferentially to V2a interneurons [44]. In both animals PS neurons were shown to increase the tail beat frequency [44,46,47] and decrease the intersegmental phase lag [44,47]. This observation suggests that different feedback topologies may allow to implement common functions to achieve the generation of an efficient swimming behavior [48].

Following a spinal cord transection, lampreys and eels can still generate coordinated swimming movements below the lesion site despite the loss of descending signals [46,49,50]. This suggests that sensory feedback can replace the missing RS excitability and CPG coupling to allow for the generation of rhythmic and coordinated outputs. This hypothesis was successfully tested in computational and robotic studies of the lamprey's swimming CPG network with proprioceptive [51] and/or lateral hydrodynamic force [52] sensing. Unlike lampreys and eels, salamanders do not display coordinated swimming immediately after spinal cord injuries [53,54]. This is likely due to a lower level of excitability of the spinal circuits below the lesion, that consequently require the regrowth of descending signals to restore locomotion [53,55]. Nonetheless, salamanders can regenerate their spinal cord and display swimming within 6-8 weeks after the lesion. Interestingly, sensory stimulation was shown to markedly enhance the recovery capability in spinally-transected eels [56].

Salamanders are unique animals to study how CPGs and sensory feedback circuits have evolved from fishes to mammals, and to explore the relative contribution of CPG and sensorimotor circuits during aquatic and terrestrial locomotion. Like other amphibians, salamanders appeared on Earth around 360 million years ago, sometime between the arrival of bony and jawless fishes (i.e. lamprey) and that of mammals [22,57,58]. Salamanders are close extant representatives of the first tetrapods that evolved from water to land. They swim in water moving the tail in an undulatory fashion reminiscent to lamprey's swimming with a characteristic rostrocaudally propagating traveling wave of motoneuron activity in alternation between the left and right sides, and walk on land using lateral sequence walking or trotting gaits ressembling

those described in mammals [15,59–62]. Salamanders are also interesting animals for their extraordinary regenerating abilities. Indeed they are the only tetrapods that can restore spinal circuits and their function after spinal cord lesions [54,55,63,64]. Some species of salamanders can regrow missing limbs [65] and even parts of the brain tissue [66]. Investigating how salamanders re-configure their nervous system to restore motor functions could help us better understand the motor recovery from spinal cord injury also in mammals [67,68].

Previous simulation studies on salamander locomotor networks have for the most part used high levels of abstraction (e.g., abstract oscillators) for the modeling of the CPG dynamics [69–72]. These models allow one to investigate of the core principles underlying the coordination between axial segments and they are suitable for neuromechanical simulations [70, 73] and robotics studies [61,72]. On the other hand, abstract models do not allow the investigation of the core rhythmogenic components of real neural networks. Similarly, the effect of sensory feedback on these models can only be studied through abstract mathematical rules that are difficult to compare to experimental findings [74,75]. In the lamprey, many studies have focused on bridging this gap with detailed bio-physical models (e.g. Hodgkin–Huxley neurons [76]) of the locomotor networks [4,77–80]. Taking inspiration from the lamprey, similar efforts have been done also for the salamander locomotor circuits [81]. Unfortunately, these models are not suitable for an extensive use in neuromechanical simulations due to their excessive computational cost. Additionally, the high-dimensionality of these models calls for an amount of data for parameter instantiation that is not currently available for most animal models. For these reasons, several efforts have been made to use an intermediate level of abstraction to study the salamander locomotor networks [82,83]. In these models, the circuits are simulated as networks of spiking neurons modeled with reduced two-dimensional formulations including membrane potential and an adaptation variable (e.g. adaptive integrate-and-fire [84], Izhikevich [85], Fitzhugh-Nagumo [86]). These models allow to grasp the core rhythmogenic mechanisms of the locomotor networks while maintaining a low computational complexity [87]. Moreover, these formulations enable the creation of closed-loop neuromechanical simulations, in which sensory feedback is integrated into the networks in a biologically-realistic manner [12]. In [82], this approach was used to investigate the role of axial and limb proprioceptive feedback during walking gaits in salamanders. Interestingly, this work showed that sensory feedback played an important role in obtaining a coordinated lateral sequence walking gait, as well as transitions from walking to trotting. On the other hand, the work focused solely on ground locomotion and it did not investigate swimming, nor the effect of modulating the strength and/or topology of the axial PS feedback.

Indeed, despite the advancements in simulating the salamander's locomotor circuits at different levels of abstraction, a full understanding of the relative contribution of CPG and axial PS feedback has so far remained elusive. In the present work, we aim to investigate this problem with a neuromechanical model of the salamander's locomotor network (see Fig 1) based on adaptive leaky integrate-and-fire spiking neurons (i.e., with an intermediate level of abstraction [84]). The model was built upon previous studies [81,88,89] and it was refined to include new findings from salamander and other species. Additionally, the spiking neural network was coupled to a simulated salamander body with realistic physical parameters in order to provide axial PS feedback to the network and to evaluate the resulting locomotor performance (see Fig 2A). The main questions addressed by the paper can be summed up as

1  Can a spiking neural network of the salamander locomotor circuits generate both swimming and walking patterns with a unified architecture?
2  How does axial PS feedback influence the performance of the model in terms of frequency, intersegmental phase lag, forward speed and locomotor efficiency?

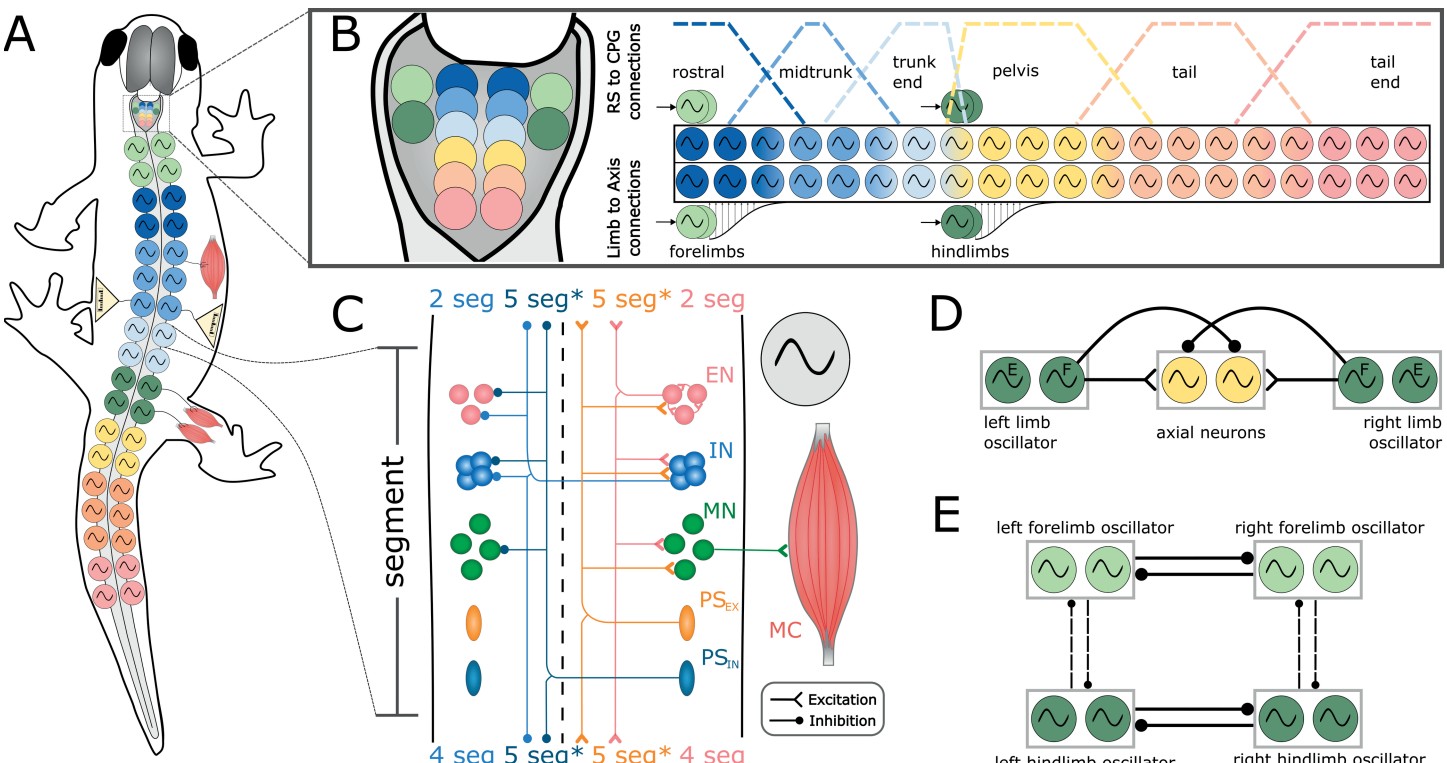

**Fig 1. Neural model. A**) shows the schematics of the locomotor network organization. The axial network is comprised of two chains of oscillators (i.e., the hemiseg-ments), divided into 40 equivalent segments (only a fraction is shown). The axial segments are spatially organized in trunk and tail sub-networks. In correspondence of the two girdles (green oscillator symbols) there are the equivalent segments for the limb networks. Each segment projects to a pair of antagonist muscle cells. The axial network receives proprioceptive sensory feedback (PS) based on the local curvature of the model. The limb network does not receive sensory feedback. **B**) shows the organization of the reticulospinal population (RS). The axial RS neurons are divided into 6 sub modules (3 for the trunk, 3 for the tail) projecting to the axial central pattern generator (CPG) neurons according to overlapping trapezoidal distributions, shown as dotted lines. The limb RS neurons are divided into 4 modules, each pro-jecting to a separate limb segment. The limb segments project to the axial CPG network with local (i.e.; spanning few segments) descending distributions. **C**) shows the organization of each equivalent segment of the CPG network. The structure comprises excitatory (EN) and inhibitory (IN) CPG neurons, motoneurons (MN), excitatory (PS_EX) and inhibitory (PS_IN) proprioceptive sensory neurons and muscle cells (MC). Only a fraction of the neurons and their connections originating from the right hemisegment are shown. The connections from the left hemisegment follow the same organization principle. The range of the ascending and descending projections is displayed in terms of equivalent segments (seg). The *PS* connection ranges (indicated with an asterisk *) were modified in the following analyses to investigate the effect of feedback topology (excitation/inhibition, ascending/descending) on the closed-loop network. **D**) shows the limb-to-body connectivity pattern. Each flexor (i.e. protractor) hemisegment of a limb oscillator excites the ipsilateral axial hemisegments and inhibits the contralateral axial hemisegments. **E**) shows the inter-limb connectivity scheme. Each limb oscillator sends inhibitory projections from the flexor side to the flexor side of the commissural and ipsilateral limbs. The commissural connections (thick lines) are stronger than the ipsilateral projections (dotted lines) to ensure the left-rialternation between opposed forelimbs and hindlimbs.

3   What is the effect of varying the topology of axial PS feedback (ascending or descend-ing, excitatory o inhibitory, local or long-range) on the rhythmic motor patterns gener-ated by the model?

4   Can axial PS feedback extend the functional range of the CPG network by enhancing its tolerance to the variability of the external drive and/or the internal parameters?

5   How does the feedback affect the model's robustness to noise and its ability to recover stable locomotor rhythms in the presence of disturbances?

The spinal neural network consists of spiking neurons of excitatory and inhibitory types that encompass different classes of neuronal populations [12]. These neuronal classes, which were genetically identified in the mouse and zebrafish [1,22], could indeed be present in sala-manders based on recent comparative genome analysis and physiological results [12,90]. The

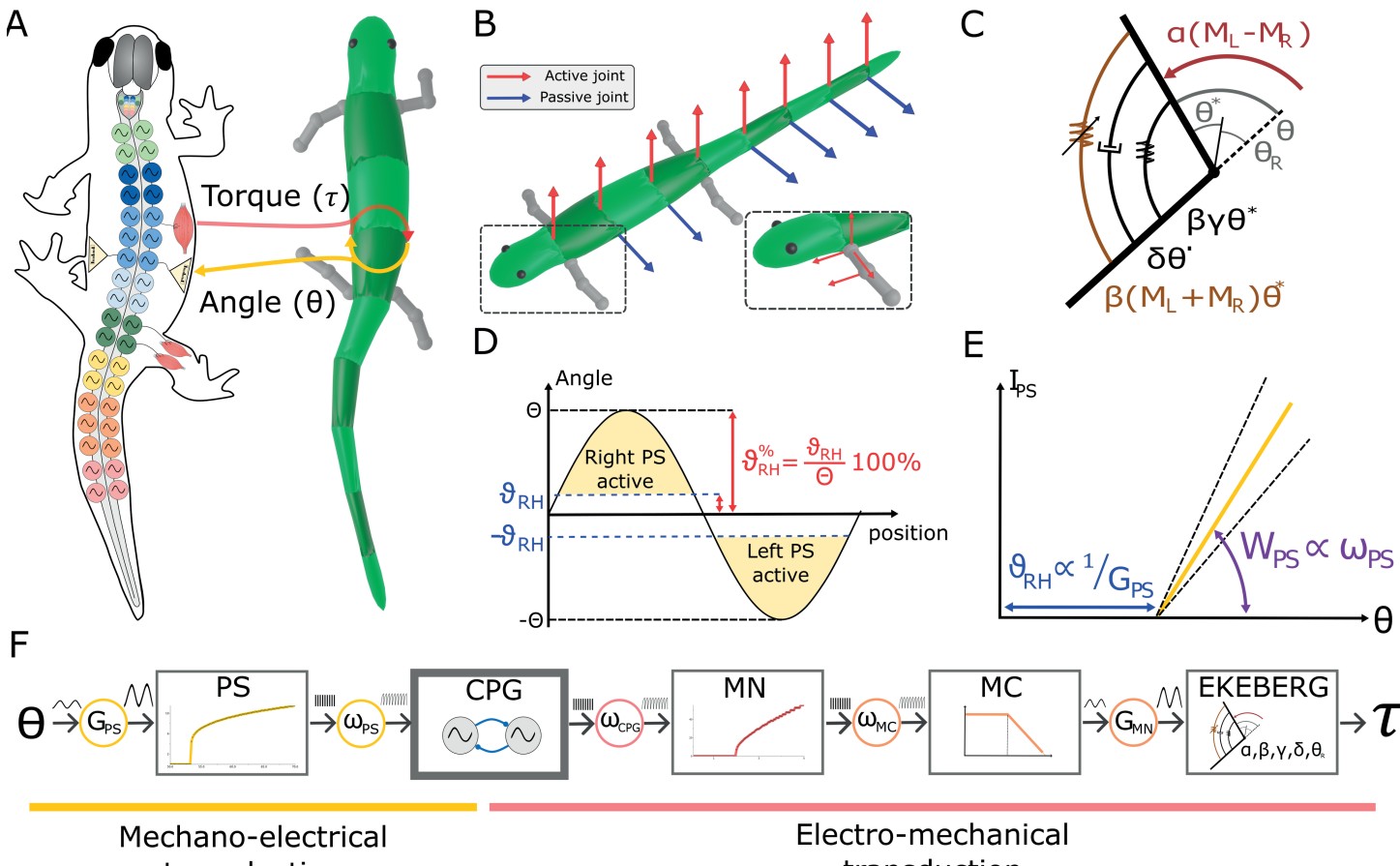

**Fig 2. Mechanical model. A)** shows the exchange of information between the neural and mechanical model. The neural model sends torque commands to the joints and reads back the corresponding joint angles. **B)** shows the active (in red) and passive (in blue) joints of the axial body. Active joints exert their action in the horizontal plane. Passive joints excert their action in the sagittal plane. The insert shows the active degrees of freedom of each limb. The shoulder and hip joints are spherical joints allowing rotations about the three axes. The elbow and knee joints are hinge joints that allow flexion/extension. **C)** displays the equivalent scheme of an Ekeberg muscle model. It is a rotational spring-damper system with the addition of a variable stiffness term $\beta(M_L + M_R)\theta^*$. The active term of the model acts an external torque $\alpha(M_L - M_R)$. **D)** shows the relationship between the joint angles ($\theta$) and the corresponding proprioceptive sensory neurons activity (PS). When $\theta$ crosses a threshold value $\theta_{RH}$ (i.e.; the rheobase angle) the PS neurons on the stretched side become active (yellow area). Notably, the ratio (in percentage) between each joint's oscillation amplitude ($\Theta$) and the corresponding threshold $\theta_{RH}$ is constant along the entire axial network and it is equal to $\theta_{RH}^{\%}$ . **E)** shows the equivalent scheme for the mechano-electrical transduction principle, in analogy with proportional control. The rheobase angle $\theta_{RH}$ indicates the critical angle for which the PS neurons become active. The feedback synaptic weight $\omega_{PS}$ modulates the strength W (i.e., the slope) of the PS action once $\theta_{RH}$ is crossed. **F)** displays the control diagram for the mechano-electrical and electro-mechanical transduction. The axial central pattern generators (CPGs) receive curvature information through the PS population. The activity of the CPGs is translated into torque information through the Ekeberg muscle model via motoneurons (MN) and muscle cells (MC).

topology of the synaptic connectivity was constrained using indirect evidence from neuronal and kinematic recordings of *Pleurodeles waltl*, and by taking inspiration from that of the lamprey and zebrafish [12,22,81]. The 3D mechanical salamander model (see Fig 2A and 2B) consists of multiple connected rigid links that faithfully resemble the body of the real salamander and it is actuated by virtual muscles [91] (see Fig 2C). Physical and muscles parameters were optimised to match the swimming kinematics of *Pleurodeles waltl* salamanders using a novel system design based on the transfer function of the linearized muscle model response (more details in S3 File). For validation, the kinematics reported from *Pleurodeles waltl* in [61,92] was replicated and matched (see S4 File). The axial PS neurons received input from the local

curvature of the simulated salamander body and they exerted direct mono-synaptic effects onto the axial CPG and motoneurons (MN) [44,45].

The rhythmogenic properties of the network were investigated by simulating fictive locomotion experiments (see section "Simulating fictive locomotion" in S5 File). The network was able to generate oscillations at the level of a full cord, full hemicord, individual segments and individual hemisegments (see Fig A in S5 File), aligning with experimental results [93,94]. When controlled by descending RS drive in open loop (i.e.; without PS feedback), the model could generate both swimming and walking gait patterns. The former pattern was obtained with a selective activation of the axial RS neurons. The latter was obtained by stimulating the limbs RS neurons while simultaneously reducing the excitation to the axial RS neurons compared to swimming. Interestingly, this is the first spiking neuromechanical model of the salamander to display both gaits with a unified network architecture (see question 1). The open loop network was also systematically studied to characterize the effects of modulating drive, noise and commissural inhibition, providing a solid basis for the understanding of the model (see section "Open loop swimming" in S5 File). In closed loop, intermediate values of stretch feedback strength boosted the salamander's frequency, speed and locomotion efficiency with respect to the pure open loop paradigm, therefore improving the swimming performance (see question 2). On the other hand, extreme levels of sensory feedback strength led to a loss in controllability of the network (i.e.; reduced range of frequencies, phase lags and speeds) and a switch to feedback-dominated rhythm generation. The study also investigated different biologically-realistic feedback topologies, namely ascending/descending and ipsilateral-excitatory/commissural-inhibitory connectivity patterns. Interestingly, the effect of stretch feedback was largely conserved across the considered connectivities and it consistently led to an increase of the frequency and a decrease of the intersegmental phase lag of the CPG oscillations (see question 3). Additionally, we proposed a mechanistic explanation of the reported sensory feedback effect and its relationship to feedback topology. The presence of sensory feedback also largely expanded the stability region of the closed-loop network with respect to the level of external stimulation and strength of commissural inhibition (see question 4). Moreover, the closed loop network was able to revert pathological activation patterns, leading to a recovery of swimming behaviors even in the presence of high levels of external noise (question 5). Overall, the proposed model helped in shining a light on the interplay between central and peripheral pattern generation in the salamander locomotor networks.

## Results and discussion

### Open loop network

**Swimming network.** The locomotor network underlying swimming behaviors can be dynamically activated through the targeted stimulation of the RS populations projecting to the axial CPG network (see Fig 3A). Upon stimulation, the RS convey a descending excitatory signal that initiates rhythmic motor patterns. The ensuing activity within the axial CPGs manifests as a left-right alternating travelling wave of neuronal activity which closely resembles the motor patterns observed both in-vitro and in-vivo in salamanders (see Fig 3Ai). The swimming activity generated by the open-loop network when linked to the mechanical model is displayed in S2 Video.

The frequency ($f_{neur}$) of the generated motor patterns can be precisely modulated by adjusting the level of external drive to the RS neurons (see Fig 3Aii). Moreover, the total wave lag (*TWL*, see Eq 3 in S1 File) of the network can be modulated by adjusting the relative drive to the rostral or caudal components of the network. Specifically, augmenting the drive to the rostral RS tends to increase the *TWL* (see Fig 3Aiii). Conversely, reducing the drive to these

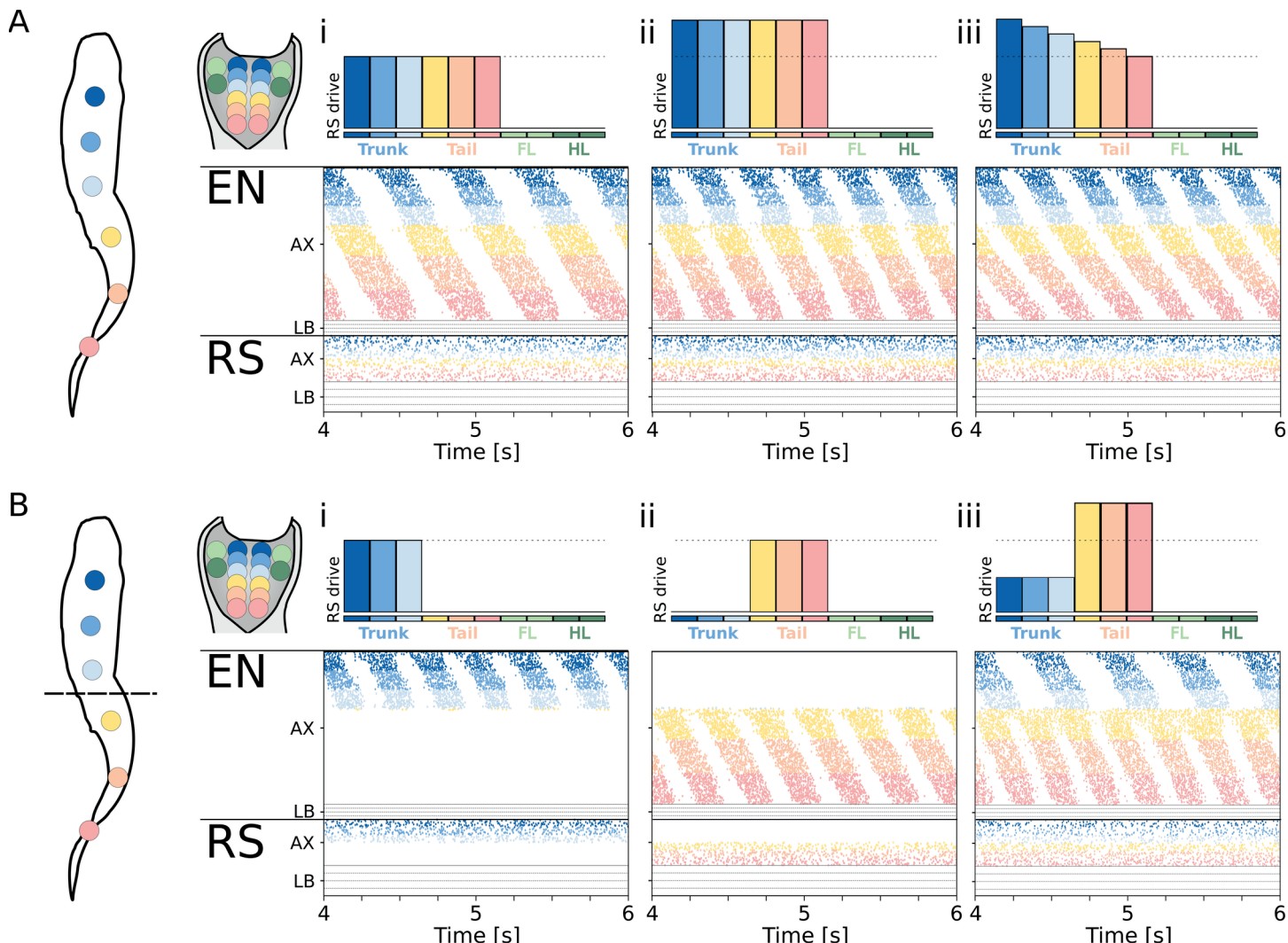

**Fig 3. Open loop swimming network. A**) shows the swimming pattern emerging from three different levels of stimulation to the axial RS. The level of stimulation provided to the 10 RS modules (6 axial, 4 limbs) is shown on top of each plot. The raster plots represent the spike times of excitatory neurons (EN) and reticulospinal neurons (RS). For the axial network (AX), only the activity of the left side is displayed. For the limb network (LB), only the activity of the flexor side is displayed. Limbs are ordered as left forelimb (LF), right forelimb (RF), left hindlimb (LH), right hindlimb (RH), from top to bottom. Ai shows the pattern obtained with a constant drive of 5.0pA. The neural activity corresponds to a frequency $f_{neur}$ = 2.1 $Hz$ and a total wave lag $TWL$ = 1.04. Aii shows the response to a higher constant drive of 5.7pA. The neural activity corresponds to $f_{neur}$ = 3.1 $Hz$ and $TWL$ = 1.27. Aiii shows the effect of driving the network with a gradient of excitation ranging from 5.7pA (to the rostral RS neurons) to 5.0pA (to the tail RS neurons). The neural activity corresponds to $f_{neur}$ = 3.1 $Hz$ and $TWL$ = 1.83. **B**) shows the coordination patterns achieved for different levels of stimulation to the trunk and tail regions. Bi shows the activity corresponding to a stimulation of 5.7pA of the trunk RS. The trunk network is rhythmically active, but the activity does not spread to the tail network. Bii shows the activity corresponding to a stimulation of 5.7pA of the tail RS. The tail network is rhythmically active, but the activity does not spread to the trunk network. Biii shows the activity corresponding to a different level stimulation of the trunk (5.0pA) and tail (6.2pA) RS neurons. Both regions are rhythmically active and the tail network oscillates at a higher frequency with a 2:1 ratio with respect to the trunk network.

neurons (or increasing the drive to caudal-projecting ones) results in a reduction of the *TWL*. This adjustment of the delay in phase synchronization supports the theoretical basis proposed by the trailing oscillator hypothesis [95]. It emphasizes the intricate control over the rhythm and coordination enabled by the modular RS drive [96].

Interestingly, the trunk and tail networks could also be independently controlled thanks to the modular RS organization and the discontinuity within the CPG connections in correspondence of the hip girdle. A targeted stimulation of the trunk RS neurons leads to the generation of a travelling wave of activity in the trunk CPG neurons, while the tail network remains silent (see Fig 3Bi). Similarly, stimulating the tail RS neurons leads to a rhythmic activity isolated to the tail network (see Fig 3Bii). The trunk and tail networks could also be simultaneously activated while displaying non-synchronized oscillations. In Fig 3Biii, an increased drive to the tail RS population leads to higher-frequency oscillations within the tail CPG network with a 2:1 ratio with respect to the trunk section. The independent control of the trunk and tail sections aligns with experimental findings from the salamander locomotor networks [97].

Notably, biological experiments often by-pass the descending brain drive to directly investigate the capability of the spinal circuits to generate locomotor activity. Such experiments typically involve the use of N-methyl-D-aspartate (NMDA) baths to increase the excitability of isolated spinal cord sections [93,94]. This configuration, denoted "fictive locomotion", was successfully tested in the current model by directly stimulating sections of the axial CPG neurons with an external current (see section "Simulating fictive locomotion" in S5 File). These experiments demonstrate the capability of the network to generate oscillations at the level of the full cord, full hemicord, isolated segments and isolated hemisegments (see Fig A in S5 File). The results show the distributed nature of the rhythmogenic capabilities of the model, in accordance with experimental results with salamanders [93,94].

Overall, the modular RS drives represent a straightforward yet potent mechanism to finely control the axial CPGs by selectively activating different sub-networks (e.g., trunk and tail) and by modulating the frequency and phase lag of the generated oscillations. This level of control underscores the integral role that the reticulospinal pathways could have in tuning the spatiotemporal dynamics of locomotor networks, facilitating adaptive motor outputs in response to varied environmental demands. Nonetheless, unless stated otherwise, the following analyses on the swimming network imposed a constant level of stimulation amplitude to all the axial RS populations in order to better highlight the key parameters affecting the model performance in open and closed loop.

Several additional analyses were carried out to characterize the open loop network and further validate it with respect to experimental evidence from the salamander locomotor circuits. More details can be found in the "Open loop swimming" section in S5 File:

- In the subsection "Effect of descending drive", the capability of the descending RS drive to modulate the frequency, phase lag and stability of the network was studied. It was found that the network shows a triphasic response to the stimulation level (see Fig Bi and Bii), switching from unpatterned spiking (for low drives) to stable swimming-like oscillations (for intermediate drives) to tonic bilateral activity (for high drives). Additionally, in the stable oscillations regime the network showed a linear relationship between the descending drive and the frequency of the oscillations, resonating with experimental evidences in salamanders [37].
- In the subsection "Effect of external noise", we investigated the capability of the network to withstand external disturbances. Interestingly, the open loop network was inherently robust to remarkable levels of noise (see Fig Ci and Cii). This is likely due to the diversity in the neural parameters of the CPG population, as suggested by [4].

- In the subsection "Open loop sensitivity analysis", the network was further characterized with a sensitivity analysis. The analysis highlighted the strong sensitivity of the network to a modulation of the commissural inhibition strength within the CPG modules (see Fig Ciii).
- In the subsection "Effect of commissural inhibition", the effect of varying the commissural inhibition strength was studied in more detail. The analysis showed that inhibition not only ensures left-right alternation in the swimming network, but it also strongly modulates the periodicity, frequency and intersegmental phase lag of the generated patterns (see Fig D).

**Walking network.** Walking behaviors can be obtained in the simulated network via the activation of the RS modules projecting to the axial and limb CPGs (see Fig 4Ai). With a stimulation value of 5.0pA and 4.0pA to the limbs and axial RS populations, respectively, the four equivalent segments displayed rhythmic activation with a frequency of 0.8 Hz (see Fig 4Aiii). Notably, the obtained frequency is lower than the ones obtained for swimming, in accordance with experimental observations [59,61]. Also, the stimulation amplitude used to achieve this stepping pattern was markedly lower than the one used for swimming, matching experiments involving MLR stimulation in salamanders [37]. The coordination between the limb oscillators is obtained through the inter-limb connectivity. The connectivity enforces the alternation of contralateral limbs via inhibitory commissural inter-limb projections. Additionally, ipsilateral limbs also display anti-phase relationship due to the presence of inhibitory ipsilateral inter-limb connections. These two conditions enforce the simultaneous activation of diagonally opposed limbs. Additionally, given the lower intrinsic frequency of the limb oscillators and their connections to the axial network, the oscillations of the axial CPGs are slowed down and they synchronize to the limb activity. Therefore, the axial network is entrained by the limb oscillators and it displays a phase shift in correspondence with the hip girdle. This pattern of activation resembles the one displayed by salamanders during trotting [61,72].

Interestingly, the trunk network displayed an average intersegmental phase lag (*IPL*, see Eq 2 in S1 File) of 0.33%, corresponding to a (quasi-) standing wave of activity. Contrarily, the tail network displayed an *IPL* of 1.62%, corresponding to a travelling wave. This observation contrasts with experimental results from behaving salamanders, where the tail segments also displayed a quasi-synchronous behavior [59,98], although with a high level of variability [99]. On the other hand, fictive locomotion experiments showed that the tail network could only display travelling waves of activation [97] when isolated from the limbs and trunk. Previous simulation studies proposed long-range limb-to-body connections to achieve a standing wave in the entire network [69]. In these models, the forelimb CPGs were connected to the entire trunk network, while the hindlimb CPGs were connected to the entire tail network [69,82] . On the other hand, these long-range projections were subsequently deemed unlikely based on the observation that limb oscillations could coexist with swimming-like patterns of axial activation during underwater stepping [72,100]. Notably, the coexistence of a traveling wave in the trunk ventral roots and rhythmic activity in the limb nerves was also the most common pattern observed during fictive locomotion in [94]. Sensory feedback could help resolve the discrepancy between fictive and in-vivo observations, providing a mechanism to synchronize the activity of the trunk and tail oscillators while maintaining local connections (i.e., limited to the proximal segments) between the limbs and the body CPGs.

Similarly to the swimming network, increasing the stimulation to the limbs RS populations to 5.5pA (while maintaining the axial RS drive of 4.0pA) lead to an increase in the oscillation

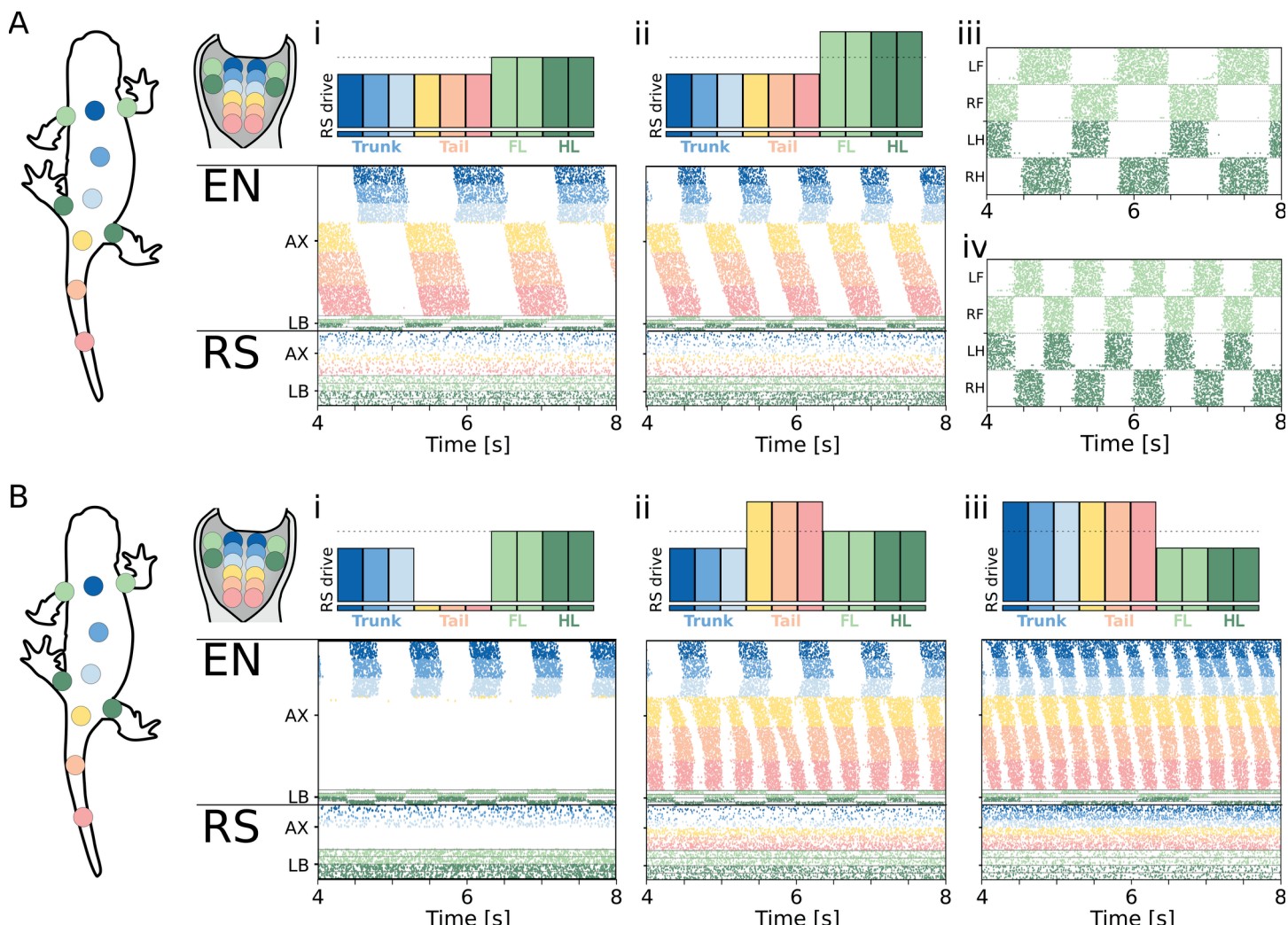

**Fig 4. Open loop walking network. A**) shows the walking pattern emerging from different levels of stimulation to the axial and limb RS. The level of stimulation provided to the 10 RS modules (6 axial, 4 limbs) is shown on top of each plot. The raster plots represent the spike times of every neuron of the spinal cord model. For the axial network (AX), only the activity of the left side is displayed. For the limb network (LB), only the activity of the flexor side is displayed. Limbs are ordered as left forelimb (LF), right forelimb (RF), left hindlimb (LH), right hindlimb (RH), from top to bottom. Ai shows the pattern obtained with a constant drive to the limbs and axial RS with an amplitude of 5.0pA and 4.0pA, respectively. The neural activity corresponds to $f_{neur}$ = 0.8 $Hz$. Aii shows the response to a higher constant drive of 5.5pA to the limb RS. The neural activity corresponds to $f_{neur}$ = 1.3 $Hz$. Aiii and Aiv show the limb activity recorded during the experiments in Ai and Aii, respectively. **B**) shows the coordination patterns achieved when modulating the drive to the axial RS modules. Bi shows the response of the network from Aii when the tail RS modules do not receive any drive. The tail network remains silent. Bii shows the response of the network from Aii when the tail RS modules receive an increased drive of 6.5pA. The tail network oscillates at a higher frequency with a 2:1 ratio with respect to the trunk network. Biii shows the response of the network with an axial RS drive of 6.0pA and a limbs RS drive of 4.5pA. The axial network displays a travelling wave of activity despite the rhythmic limb activation.

frequency to 1.3 Hz (see Fig 4Aii), while maintaining the synchronization between diagonally opposed limbs (see Fig 4Aiv). In principle, different connectivity patterns between the limbs could be achieved by modulating the inter-limb connectivity and/or the descending drive from the RS neurons [39,101,102]. Nonetheless, the study of additional walking patterns was outside of the scope of this work and it was not investigated further.

Analogously to the swimming network, the differential stimulation to the trunk and tail RS populations could be exploited to further increase the flexibility of the model. Silencing

the drive to the tail RS neurons led to oscillations in the trunk and limb segments that did not spread to the tail CPG section (see Fig 4Bi). This pattern corresponds to stepping with a passive tail, a behavior that is observed in behaving salamanders [72,100]. Contrarily, increasing the stimulation to the tail RS populations could lead to a "double-bursting" in the tail section (see Fig 4Bii). This pattern was observed in-vivo in salamanders walking on slippery surfaces [59,99]. Finally, the activation of the limb CPGs did not necessarily result in a 180* phase shift between the trunk and tail oscillations. Indeed, increasing the drive to all the axial RS neurons allowed the axial network to escape the frequency-locking imposed by the limbs (see Fig 4Biii). In this case, the network displayed rhythmic limb activity coexisting with a full travelling wave of axial activity, analogously to the patterns observed during underwater stepping in salamanders [72,94]. As previously discussed, the coexistence of a travelling wave and limb activity is possible thanks to the local nature of the connections between the limbs and the axis.

When linked to the mechanical model, the network could reproduce a biologically-realistic trotting gait (see S3 Video). The generated walking had a frequency of 0.9 Hz and a stride length of $0.43 SVL/cycle$, which resembles kinematics data from salamanders [61]. Interestingly, this is the first spiking neuromechanical model of the salamander to display both swimming and walking gaits with a unified network architecture (see question 1). This model can therefore be the basis for investigating both locomotor modes while maintaining biologically-realisitc implementations of motor control, sensing and central pattern generation. On the other hand, the use of a single equivalent segment for each limb limits the capability of the model to replicate the complex inter-joint coordination patterns displayed by each limb during ground locomotion. Additionally, the necessary speed-dependent change in the amplitude of the angles of axial and limbs joints could not be reproduced with constant motor output gains (see $G_{MC}$ in Eq 10). Finally, the current network design does not allow to modulate the duty cycle of the flexion-extension phases of each limb, which were forced to 50% of the period. Overall, these limitations lead to difficulty in controlling the mechanical stability of the model during ground locomotion. For this reason, the current work focused primarily on investigating the effect of axial proprioceptive feedback on the swimming locomotor networks of the salamander. Future studies will improve on the aforementioned limitations to better understand the role that sensory feedback plays during walking behaviors.

## Closed loop network

**The effect of axial sensory feedback is largely consistent across different feedback topologies.** Stretch feedback serves the purpose of responding to a high level of bending on one side of the body with an activation of the muscles that control the contracted side. This mechanism is achieved in animals through the excitation of the locomotor circuits on the stretched side and/or through the inhibition of the locomotor circuits on the contracting side.

Different animals were shown to display different patterns of effects and connectivities between the stretch-sensitive neurons and the locomotor network. In the lamprey, both ipsilateral excitatory neurons and inhibitory commissural neurons were reported [42,43,45]. These sensory neurons, called Edge Cells, were shown to lead to an increase in the frequency and a decrease in the intersegmental phase lag of the generated axial oscillations [47]. In larval zebrafish, two classes of mechanosensory neurons were found to have an active role during locomotion [103]. The first class includes ipsilaterally-projecting Glutamatergic Rohon-Beard (RB) neurons. These neurons are sensitive to touch stimuli and they were shown to degenerate over the first few weeks of development [104]. The second class includes GABAergic cerebrospinal fluid-contacting (CSF-c) neurons. These neurons are sensitive to body bending and

many other chemical and neuronal cues [105,106]. Interestingly, CSF-c neurons were shown to respond to a compression (rather than stretching) of the spinal cord by inhibiting the ipsilateral CPGs and motor neurons [105]. In the adult zebrafish, the intraspinal lateral proprioceptor organ (ILP) is the only reported class of intraspinal stretch-sensitive neuron. The ILP organs were shown to have commissural inhibitory projections to pre-motor interneurons [44]. Similarly to the Edge Cells in the lamprey, these neurons were shown to increase the frequency and decrease the intersegmental phase lag of the axial CPG activity [44].

Interestingly, the described sensory feedback connections typically present an asymmetry between caudally-oriented connections and rostrally-oriented connections. Both in the zebrafish and in the lamprey, stretch-sensitive organs were reported to connect preferentially to neurons located more rostrally in the spinal cord [41,44]. This variety of solutions found in animals raises the question about what is the effect of sensory feedback connectivity on the locomotor networks and whether there are different effects related to the different types of feedback (see question 3).

For this reason, we studied the effect of feedback topology by varying the type of feedback type (ipsilateral excitation or contralateral inhibition) and the range of ascending and/or descending connections to the locomotor circuits (see Fig 5). The analysis included four distinct connectivity schemes, namely ascending ipsilateral excitation (EX-UP), descending ipsilateral excitation (EX-DW), ascending contralateral inhibition (IN-UP) and descending contralateral inhibition (IN-DW). The range of the connections (PS range) was varied in the range $[0.0, 2.5] cm$ (i.e., between 0 and 10 equivalent segments). Similarly, the PS synaptic weight ($\omega_{PS}$) was varied in the range $[0, 5]$.

Studying the effect on the frequency ($f_{neur}$) it was observed that increasing the feedback weight generally led to an increase of the frequency of oscillations (see Fig 5A). The effect was stronger for ascending (UP) connections with respect to descending connections (DW). Moreover, the effect was also stronger for excitatory projections (EX) with respect to inhibitory projections (IN). With respect to the length of the projections, a higher range generally corresponded to a higher oscillation frequency. The relationship was more nonlinear for IN-UP projections, although the highest frequency was still observed in the network with the highest PS weight and range. Overall the trend observed for the frequency was largely conserved across all the considered connection types (see question 3). In the limit case ($PS_{range}$ = 2.5$mm$, $\omega_{PS}$ = 5), all the topologies lead to an increase of $f_{neur}$ compared to the open-loop case. The relative increase was equal to +41% (EX-UP), +36% (EX-DW), +36% (IN-UP) and +22% (IN-DW), respectively. This observation suggests that the result is a general consequence of introducing stretch feedback in the locomotor circuits (see question 2). Notably, increasing the strength of the inhibitory PS connections had an opposite effect on $f_{neur}$ with respect to increasing the commissural CPG inhibition strength (see Fig D and corresponding section in S5 File), suggesting that the two connection types play different roles in the locomotor circuits. A mechanistic explanation of this observation is provided in the next section.

Studying the effect on the total wave lag (*TWL*, see Eq 3 in S1 File) it was observed that, irrespective of the feedback topology, long range connections with high values of feedback weight lead to a decrease of *TWL* (see Fig 5B). Descending connections had a markedly higher effect, leading to lower values *TWL* that correspond to quasi-standing waves of neural activity (i.e., *TWL*≈ 0). Additionally, for descending connections, *TWL* decreased monotonically both for an increase of the feedback weight and for an increase of the feedback range. Conversely, ascending connections with low or intermediate values of feedback weight, almost did not affect the value of *TWL* with respect to the open loop one. A decrease in the

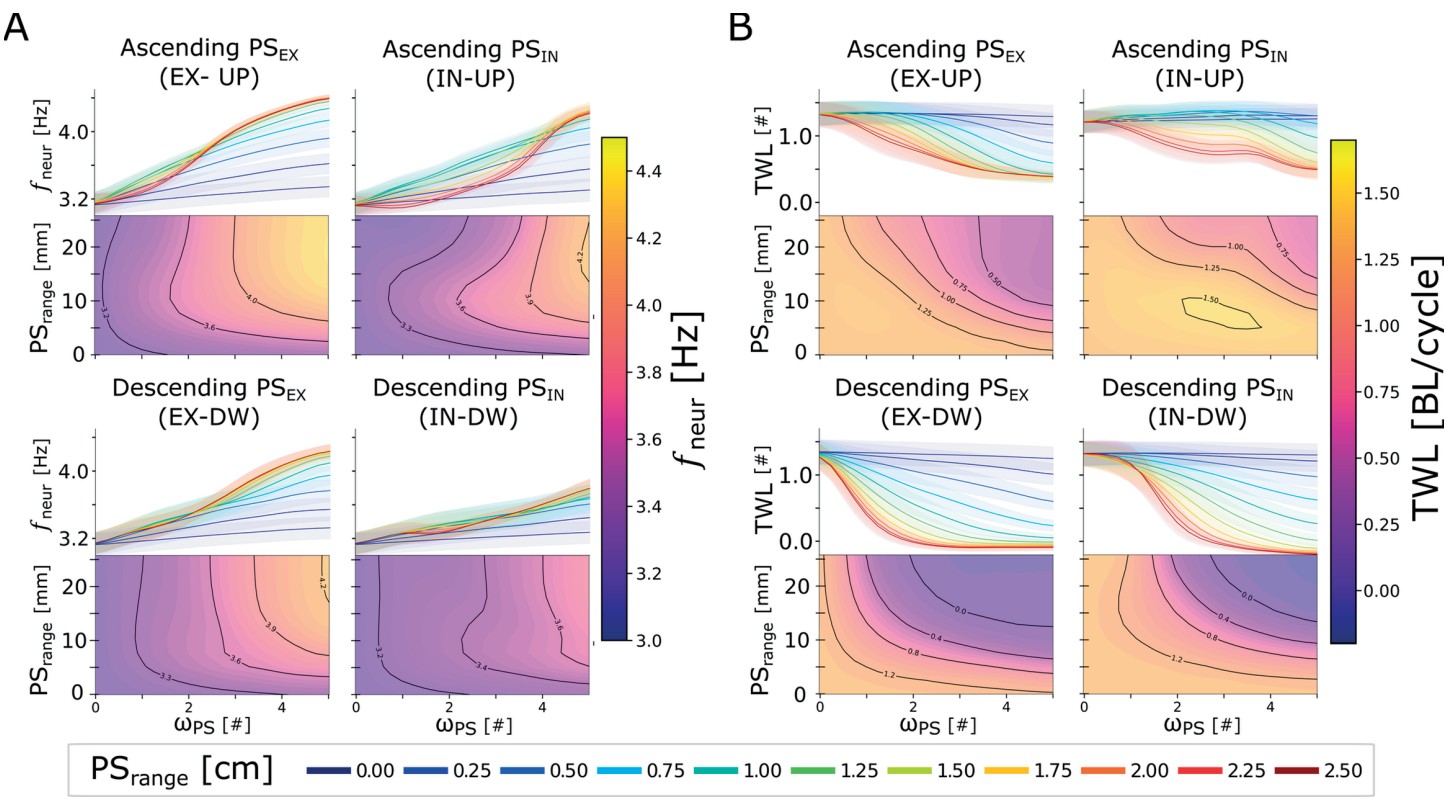

**Fig 5. Frequency and phase lag modulation during swimming with different sensory feedback topologies and strengths.** Panels A and B show the effect of different PS topology patterns and weights on the frequency ($f_{neur}$) and total wave lag ($TWL$) of the network's activity, respectively. The studied topologies include ascending excitation (EX-UP), descending excitation (EX-DW), ascending inhibition (IN-UP) and descending inhibition (IN-DW). The range of the connections (PS range) was varied in the range $[0, 2.5]\,mm$ (i.e., between 0 and 10 equivalent segments). Similarly, the PS synaptic weight ($\omega_{PS}$) was varied in the range $[0, 5]$. The rheobase angle ratio was set to $\theta_{RH}^{\%} = 10\%$. **A)** shows the modulation of the oscillations' frequency ($f_{neur}$). In each quadrant, the contour plot shows the dependency of the $f_{neur}$ on PS range and $\omega_{PS}$. On top, the projection of the relationship in the $f_{neur} - \omega_{PS}$ plane is displayed to better highlight the asymptotical values reached by $f_{neur}$. The different curves represent the frequency expressed by the network with different PS connection ranges (i.e.; horizontal cuts of the contour plot), whose value is reported in the legend. The thick lines represent the mean metric values across 20 instances of the network, built with different seeds for the random number generator. The shaded areas represent one standard deviation of difference from the mean. **B)** follows the same organization as the previous panel to represent the effect of feedback topology on the neural total wave lag ($TWL$).

$TWL$ value was only observed for long range ascending connections with high value of feedback weight. In the limit case ($PS_{range}$ = 2.5 mm, $\omega_{PS}$ = 5), all the topologies lead to a large decrease of $TWL$ compared to the open-loop case. The relative decrease was equal to −73% (EX-UP), −108% (EX-DW), −59% (IN-UP) and −115% (IN-DW), respectively. Notably, the reduction in the TWL expressed by the network is consistent with the observation that in-vitro intersegmental phase lags are distributed over a larger range of values with respect to in-vivo preparations [94]. A mechanistic explanation of this observation is provided in the next section.

The results of the analyses showed similar trends when changing the rheobase angle ratio ($\theta_{RH}^{\%}$, see Eq 14) from 10% to 50% (see Fig G in S5 File). In particular, the frequency of the oscillations was only slightly increased (up to 14%) regardless of the feedback topology (see Fig Gi). Conversely, the effect on the total wave lag was comparable to the one described for $\theta_{RH}^{\%}$ set to 10% (see Fig Gii). Indeed, in the limit case, the TWL was reduced by −58% (EX-UP), −98% (EX-DW), −53% (IN-UP) and −95% (IN-DW), in agreement with the previous analysis.

Overall, the results show that $f_{neur}$ is strongly dependent both on the critical angle of activation $\theta_{RH}^{\%}$ of the sensory neurons and on the weight of their synaptic connections $\omega_{PS}$. On the other hand, the total wave lag was primarily affected by $\omega_{PS}$ irrespective of the choice of $\theta_{RH}^{\%}$ (see question 2). While in the extreme scenarios (i.e. high $\omega_{PS}$ and $PS_{range}$) we found a high degree of invariance between the different connection types, there were notable differences in the PS effects for the intermediate cases. Indeed, the range of descending connections affected the total wave lag to a much larger extent with respect to ascending connections. On the other hand, the frequency followed a similar trend irrespective of the considered topology. This observation could provide an explanation for the asymmetry found in the connectivity of sensory feedback organs of swimming animals between ascending and descending projections (see question 3).

Notably, this analysis investigated only a fraction of the possible feedback topologies, namely the ones involving a single connection type (ipsilateral EX or commissural IN) projecting continuously to rostrally or caudally-located target neurons. The effect of feedback could vary when considering multiple coexisting PS types (e.g. EX-UP and IN-UP), discrete connectivity schemes (e.g. targeting segments at a specific distance as in [52]) or different target populations (e.g. PS to PS connections, as reported in [44]). Given the absence of anatomical data regarding the connectivity of the sensory neurons within the spinal cord of salamanders, the analyses in the next sections were performed using symmetrical ascending and descending PS projections and a range of 5 segments, a conservative hypothesis that is consistent with the range observed in the zebrafish and lamprey [44,45].

**The neuromechanical principles of frequency and phase-lag control by stretch feedback during swimming.** In Fig 5 numerical simulations demonstrated that increasing the stretch feedback $w_{PS}$ led to a global increase in the frequency and to a decrease in the phase lag of the oscillatory traveling waves. In this section, we provide some mechanistic arguments for these behaviors (see question 3). In particular, we aim to explain why (1) the frequency of the network increases with stronger feedback strengths and (2) the wave lag of the closed loop network with ascending PS projections (both $PS_{IN}$ and $PS_{EX}$) settles to a positive value while descending PS projections force the network to a quasi standing wave (as shown numerically in Fig 5B, $TWL \approx 0$ for EX-DW and IN-DW).

During locomotion the body bends on one side following the activation of the CPG neurons on the ipsilateral (same) side (see Fig 6Ai). After some delay, the curvature crosses the rheobase angle $\theta_{RH}$, which in turn activates PS neurons on the opposite side, according to the mechano-electrical transduction relationship (see Eq 13). During the time interval when the body curvature is higher than $\theta_{RH}$, $PS_{IN}$ neurons inhibit the CPGs on the opposite side, while $PS_{EX}$ excite the CPGs on the same side. The former interaction facilitates the burst termination of the (active) opposite-sided CPGs, while the latter facilitates the burst initiation of the (inactive) same-sided CPGs. This mechanism causes an anticipated switch in the CPG activity, in accordance with experimental results on lampreys [43,107]. The effect is analogous to the action of a bilinear torsional spring that acts to restore the straight position of the body once a critical angle is crossed (see Fig 6Aii). The activation point of the spring and its elastic constant are equivalent to the rheobase angle $\theta_{RH}$ and slope $W_{PS}$ of the PS's mechanotransduction relationship, respectively (see Fig 2E). This theoretical analogy confirms the increase in the swimming frequency with increasing $w_{PS} \sim W_{PS}$ and/or decreasing $\theta_{RH}^{\%}$ values predicted numerically (see Fig 5A and text).

In terms of the impact on the total wave lag, one can assess the influence of global connections from $PS_{IN}$ to CPG, as shown by the illustrative diagram in Figure 6B. Without sensory feedback, the CPG network produces an oscillatory traveling wave propagating with positive

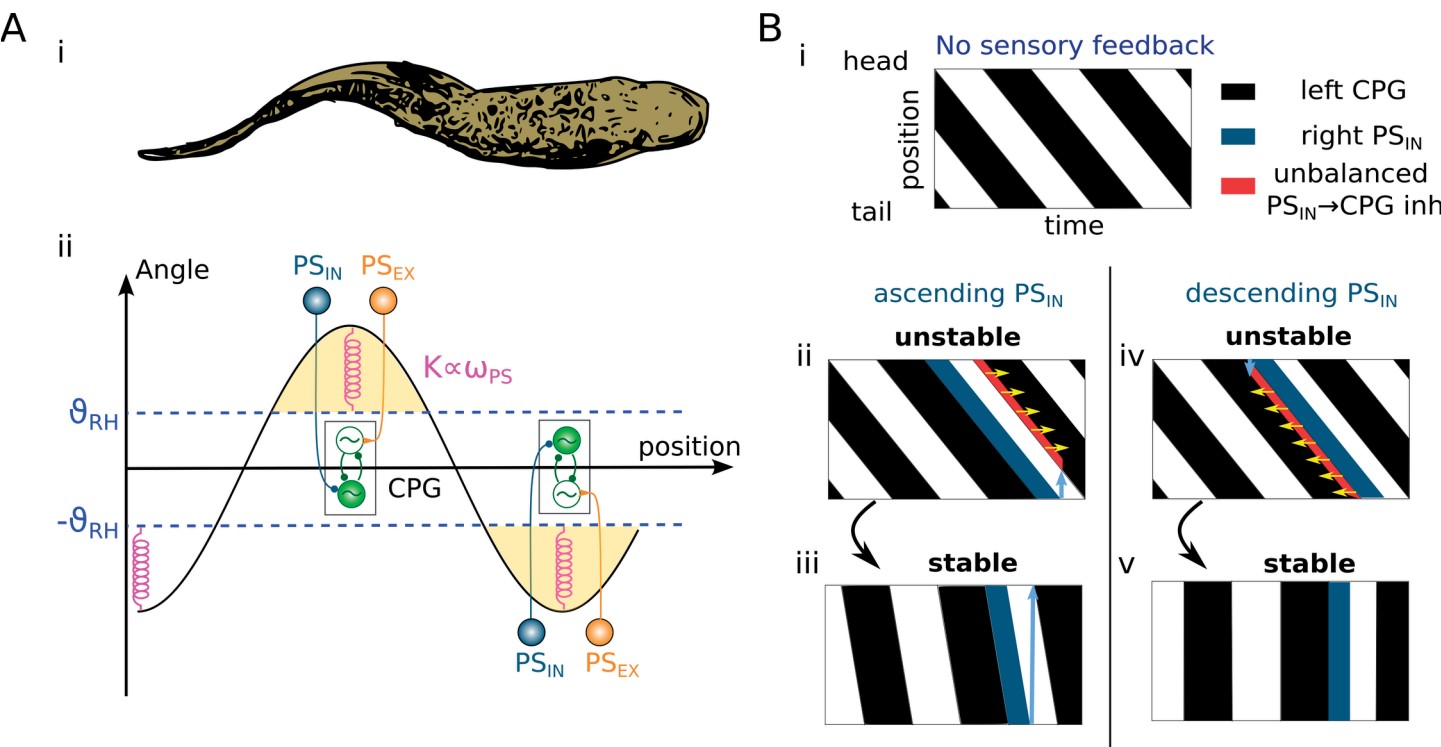

**Fig 6. Illustrative diagrams showing the mechanism of frequency and phase lag control by stretch feedback.** **A**) i shows a sketch of a sinusoidal-like body curvature during swimming. Aii shows populations of CPG and PS neurons at the location of maximal curvatures and their connections (active population have filled colors, inactive populations have white inner colors). The activity of PS neurons in correspondence of large curvatures (above $\theta_{RH}$) acts as an equivalent spring that induces an increase of the swimming frequency. The figure is drawn for $\theta_{RH}^{\%} = 40\%$ to better highlight the components of the network. **B**) shows the stereotypical CPG activities during swimming in a brief time window (black=active/firing, white=inactive) sorted according to the CPG positions (top=head positions, bottom=tail positions). Bi shows the activities of CPGs with no feedback (in open loop). Bii-Bv also show the activities of CPGs and left proprioceptive $PS_{IN}$ neurons (dark blue) in closed loop. The sub-panels Bii-Biii and Biv-Bv show subsequent time-windows at starting and final stages of activities from the time when inhibitory proprioceptive sensory feedback ($PS_{IN}$) is turned on. The light blue arrows represent the $PS_{IN}$ projections in the ascending (Bii-Biii) and descending (Biv-v) directions to the closest CPG neurons on the opposite sides, respectively. These indicate the end/start of a region of unbalanced (i.e.; not uniformly distributed across the network) inhibition from $PS_{IN}$ to the CPGs (red areas). This imbalance in $PS_{IN}$ inhibition causes a delay shift in the activations of the corresponding CPGs according to the direction indicated by the yellow arrows. See the text for more details.

wave lag (Fig 6Bi, black = active left CPG), similar to the wave shown numerically in Fig 3Ai. When feedback is introduced, the PS neurons actively contribute to the switch in the activity of the left-right CPGs, generating an imbalanced longitudinal inhibition that destabilizes the wave propagation (red areas in Fig 6Bii and 6Biv).

In the case of ascending $PS_{IN}$ (Fig 6Bii), rostral CPGs receive inhibition from $PS_{IN}$ activity, while caudal CPGs (starting from the light blue arrow) are not inhibited. This imbalance makes this configuration unstable for the closed loop network. The additional $PS_{IN}$ inhibition delays the activation of the rostral segments (yellow arrows), thus reducing the overall wave lag. The network will therefore evolve to reach a stable state where $PS_{IN}$ do not interact with the CPGs at their activation times (i.e.; the blue arrow in Fig 6Biii does not intercept the next CPG wave). In this configuration the network has minimized the intersection area between the PS activity and the onset of CPG activity (i.e., the red area). Notably, the stability is reached for a positive phase lag value where the natural tendency of the open loop network to produce a rostro-caudal wave is balanced by the synchronizing action of PS feedback. This is in accordance with the numerical results in Fig 5B (IN-UP).

In the case of descending $PS_{IN}$ the more caudal CPGs receive an additional (and stronger) $PS_{IN}$ inhibition. This additional inhibition anticipates the cycle termination in the caudal CPGs, leading to a decrease in the total wave lag (yellow arrows). In this case, however, any decrease of wave lag will cause even more longitudinal imbalance in the inhibition to the CPGs. The wave will therefore be unstable until it reaches approximately a 0 wave lag (see Fig 6Bv). This is in accordance with the numerical results in Fig 5B (IN-DW)

The same line of reasoning applies to the excitatory $PS_{EX}$ neurons, as discussed in Fig H in S5 File. These considerations lead to the same values of wave lags for the ascending and descending $PS_{IN}$ ($PS_{EX}$) connections, as found numerically in Fig 5. Interestingly, these observations explain why, in simulations, ascending projections (either excitatory or inhibitory) led to a positive $TWL$ (for both $PS_{IN}$ and $PS_{EX}$) while descending projections caused an approximately zero wave lag (see Fig 5).

Overall, the frequency increase can be explained as a local (i.e.; acting at a segmental level) spring-like effect by the stretch sensors to the CPGs in correspondence of regions with large curvature (see Fig 6A). Conversely, the decrease in total wave lag can be explained as a global (i.e.; acting at a network level) effect derived from the length of the sensory projections and the natural tendency of the network to produce rostro-caudal waves of activation (see Fig 6B).

**Intermediate levels of sensory feedback improve the swimming performance.** The effect of drive on the closed-loop network was studied by varying the value of the feedback connections weight $\omega_{PS}$. The stimulation amplitude was varied in the range [3.0, 9.0]pA. The feedback weight was varied in the range [0.0, 5.0]. For each combination of the parameters, 20 instances of the network were built with different seeds for the random number generation and the results are reported in terms of mean and standard deviation of the computed metrics (see Fig 7). The figure was obtained for a rheobase angle ratio ($\theta^{\%}_{RH}$, see Eq 14) of 10%. The analysis was also performed for $\theta^{\%}_{RH}$ = 50% and the corresponding results are reported in Fig I in S5 File. For an equivalent analysis in absence of sensory feedback, see subsection "Effect of descending drive" and Fig B in S5 file.

When studying the effect on the network's periodicity (PTCC, see Eq 4 in S1 File), increasing the value of $\omega_{PS}$ results in an expansion of the stable oscillations region with respect to the range of the input stimulation (see Fig 7A). Indeed, the range of stimulation values for which $PTCC > 1$ increased by 50% for $\omega_{PS}$ = 1.0 and it reached up to a 90% increase for $\omega_{PS}$ = 5.0. For low values of stimulation, sensory feedback allows to stabilize the spurious activations displayed by the network, giving rise to a more patterned locomotor output and a higher PTCC (see Fig 7Aii). For high values of stimulation, sensory feedback acts as an additional commissural inhibition pathway, delaying the emergence of bilateral tonic activity and, consequently, also the drop in PTCC values. Additionally, the mean PTCC value across all stimulation amplitudes increased monotonically with $\omega_{PS}$ (see Fig 7Aiii, black dots), underlining the beneficial effect of sensory feedback with respect to the stability of the oscillations. Overall, feedback can extend the functional range of the CPG network by enhancing its tolerance to the level of external stimulation (see question 4)

Interestingly, the same trends were observed for $\theta^{\%}_{RH}$ = 50% (see Fig Ii in S5 File). In this case, however, the region of stable oscillations was expanded to a lower extent (+29% for $\omega_{PS}$ = 1, +41% for $\omega_{PS}$ = 5). This decreased effect is easily understood when considering the open-loop network activity at the limits of its stability region (see Fig Bii in S5 File). For low drives (upper panel in the figure), the network produces sparse and unpatterned oscillations that do not lead to a strong muscle activation on either side of the body. For high drives (lower panel in the figure), the network produces bilateral tonic activity that results in a similar level of co-contraction for each antagonist muscle pair. In both cases, the corresponding

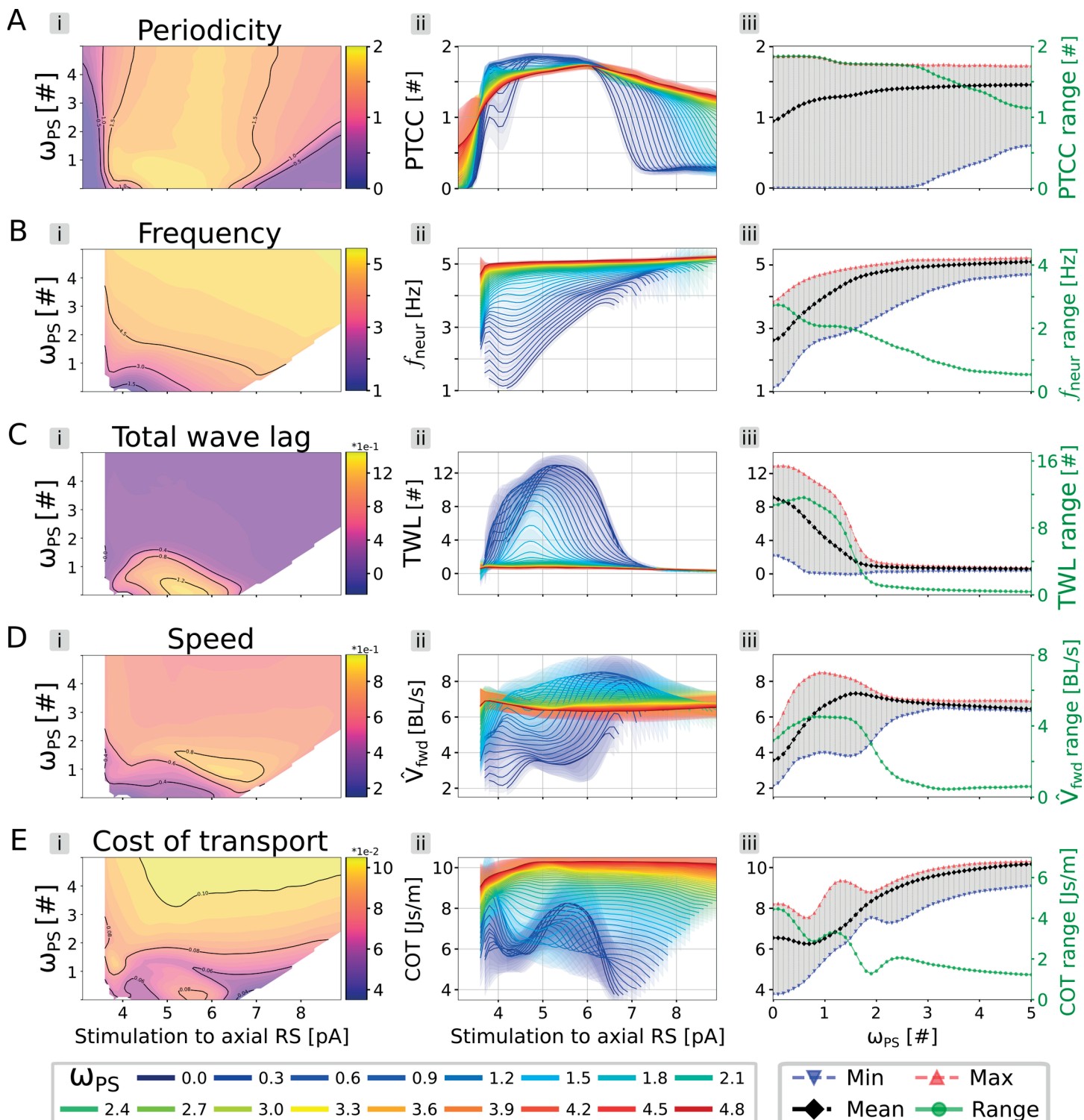

**Fig 7. Intermediate feedback strengths improve the performance of the closed loop swimming network.** In each panel, the stimulation amplitude to the axial RS neurons was varied in the range $[3.0, 9.0]$pA, while the feedback weight ($\omega_{PS}$) was varied in the range $[0.0, 5.0]$. The rheobase angle ratio was set to $\theta_{RH}^{\%} = 10\%$. The displayed data is averaged across 20 instances of the network, built with different seeds for the random number generator. In Aii-Eii, the curves differ by their corresponding value of $\omega_{PS}$, reported in the bottom legend. The shaded areas represent one standard deviation of difference from the mean. In Aiii-Eiii, the minimum (in blue), maximum (in red) average (in black) and range (in green) for the corresponding metrics across all the considered stimulation values are shown. The range values are reported on the y-axis on the right. Panels (**A-E**) display the effect on the network's periodicity ($PTCC$), frequency ($f_{neur}$), total wave lag (TWL), speed ($V_{fwd}$) and cost of transport (COT), respectively. In panels B-E, combinations with $PTCC < 1$ were not displayed.

joint angles are very low and they are less likely to reach the minimum activation angle when $\theta_{RH}^{\%}$ is increased from 10% to 50%. Consequently, increasing the value of $\theta_{RH}^{\%}$ prevents the sensory feedback from exerting its beneficial action on the locomotor rhythm, leading to a lower expansion of the stability region.

Regarding the frequency ($f_{neur}$), as observed in the previous sections, increasing $\omega_{PS}$ monotonically increases the frequency of the oscillations independently from the stimulation amplitude (see Fig 7B). Interestingly, the frequency does not increase indefinitely. Instead, $f_{neur}$ settles at a specific value of 5.23 $Hz$, acting as a feedback-related resonance frequency of the neuromechanical system. A similar saturation effect was observed for $\theta_{RH}^{\%}$ = 50% (see Fig Iii in S5 File). In this case, however, the higher rheobase angle ratio led to a lower resonant frequency of 4.20 Hz.

Notably, the range of variation of $f_{neur}$ decreased monotonically with $\omega_{PS}$, (see Fig 7Biii, green dots). This observation highlights the trade-off between robustness and flexibility of the locomotor circuits. A network dominated by sensory feedback oscillates over a wider range of input drives but it also loses the capability to modulate the frequency of the generated patterns (see question 4).

The described trade-off is even more evident when studying the effect of $\omega_{PS}$ on the total wave lag (*TWL*, see Eq 3 in S1 File) with varying values of stimulation (see Fig 7C). Indeed, for $\omega_{PS} > 2$ the network only displays quasi-standing waves of activation independently of the drive (see Fig 7Cii and 7Ciii). This further emphasizes the necessity for intermediate feedback weights that do not lead to a sensory-dominated network (see question 2).

Interestingly, the highest range of variation of *TWL* was observed for $\omega_{PS}$ = 0.7 (see Fig 7Ciii, green dots). Similarly, for $\theta_{RH}^{\%}$ = 50%, the highest range of variation was observed for $\omega_{PS}$ = 0.5 (see Fig Iiii in S5 File). These results suggest that an intermediate level of sensory feedback could increase the network's flexibility in phase lag modulation (see question 2).

With respect to the speed ($V_{fwd}$), it can be seen that increasing $\omega_{PS}$ also increases the speed of locomotion (see Fig 7D). This effect is primarily due to the augmented oscillation frequency discussed above. Concurrently, intermediate levels of $\omega_{PS}$ also lead to an increase in the average stride length (see Fig J in S5 File). Indeed, both the stride length and the speed display a local maximum for $\omega_{PS} \approx 1$ and a stimulation amplitude of 6.5 pA (see Fig 7Di and Fig Ji in S5 File). This result suggests that the model produces an optimal forward propulsion when there is a balance between the sensory feedback signal and the activity of the CPG network (see question 2). Moreover, similarly to the previous metrics, for $\omega_{PS} > 2$ we observe a decrease in the speed and stride length of the generated locomotion. Concurrently, also the range of variation of the two metrics drops markedly for high values of feedback weight (see Fig 7Diii and Fig Jiii in S5 File). Overall, these observations further show that a feedback-dominated network does not produce efficient swimming. The discussed effects were also observed when changing the rheobase angle ratio to 50% (see Fig Iiv in S5 File), increasing the reliability and of the drawn conclusions.

Finally, with respect to the cost of transport (*COT*), we observe a complex non-linear relationship with respect to the weight of the sensory feedback (see Fig 7Ei). Interestingly, there appears to be a region of minimal cost of transport in correspondence of $\omega_{PS} \approx 0.7$. Indeed, the average COT across all stimulation values displays a global minimum in correspondence of such feedback weight (see Fig 7Eiii, black dots). In that region, the simulated salamander swims more efficiently with respect to the open loop network as well as with respect to the closed loop networks with higher values of feedback weight (see question 2).

Interestingly, there is a mismatch between the $\omega_{PS}$ values that led to an optimal speed and to an optimal cost of transport. Indeed, robotics experiments proved that the kinematics necessary for efficient swimming differs from the one that optimises the speed of locomotion [108]. This observation suggests that a modulation of the sensory feedback weight could serve as a strategy for a transition between different swimming modes (e.g., cruising and escaping)

Overall, the results show that the best swimming performance is obtained with a balanced combination of open-loop and feedback-based rhythm generation (see question 2 and question 4). A purely open-loop network is more flexible in terms of frequency modulation but it lacks in terms of robustness and swimming performance. A feedback-dominated network robustly oscillates at high frequencies but is not controllable through the descending drive. Finally, balanced combinations of the two rhythm generation strategies lead to optimal swimming networks. The swimming activity generated by the closed-loop network when linked to the mechanical model is displayed in S4 Video.

**Sensory feedback extends the region of noise rejection.**   The influence of noise on locomotor patterns was examined in the closed-loop network with varying levels of $\omega_{PS}$ (see Fig 8A and 8B). The noise level was varied in the range $[3.0, 9.5]\, pA$. The feedback weight was varied in the range $[0.0, 3.0]$. For each combination of the parameters, 20 instances of the network were built with different seeds for the random number generation and the results are reported in terms of mean and standard deviation of the computed network periodicity (PTCC). For an equivalent analysis in absence of sensory feedback, see subsection "Effect of external noise" and Fig Ci and Cii in S5 File.

Fig 8A and 8B shows that sensory feedback greatly extends the stability region of the swimming network (see question 5). Indeed, for $\omega_{PS} = 0.5$, the network could tolerate noise levels 13% higher than the open loop configuration while maintaining PTCC values higher than 1. Furthermore, the improvement scaled with $\omega_{PS}$ itself (+44% for $\omega_{PS} = 1.0$, +81% for $\omega_{PS} = 1.5$). This finding supports the idea of sensory feedback serving as a redundant rhythmogenic mechanism that boosts the robustness of the locomotor networks, as mentioned in [52].

To further showcase the positive effect of sensory feedback in reducing noise interference during swimming, a simulation was carried out starting with an initially unstable open-loop network (see Fig 8C). In this scenario, the noise amplitude was set to 5.6 pA, which disrupted the swimming locomotor pattern. At time 2.5s, sensory feedback was activated with $\omega_{PS} = 0.7$. This activation revived the activity of the axial CPG network, leading to rhythmic activation of the motoneurons (see Fig 8C) and, subsequently, oscillations in the axial joints (see Fig 8D), restoring the swimming behavior. The activity switch had a transient time of approximately two seconds. During this time interval, small oscillations generated in correspondence with the head and tail joints lead to the activation of their respective PS neuronal pools. This activation ultimately spread to the entire network, resulting in a full wave of left-right alternating activity (see Fig 8C and 8D). The resulting locomotion was a forward swimming gait with kinematics that closely resembled the one produced in the absence of noise (see S5 Video and Fig 8E).

Interestingly, the same switch mechanism could be achieved also for higher noise levels, provided that $\omega_{PS}$ was set to a sufficiently large value. In Fig K in S5 File, the feedback weight was set to 2.0 with an initial noise amplitude of 8.75 pA. Despite the higher noise level, the emergence of synchronized oscillations was faster than in the previous case. On the other hand, the high $\omega_{PS}$ resulted in a quasi-standing wave of activity and, consequently, in inefficient swimming kinematics (see S6 Video).

It is important to note that sensory feedback could restore the rhythmic activity only in cases where the noisy network was capable of generating joints oscillations that could reach

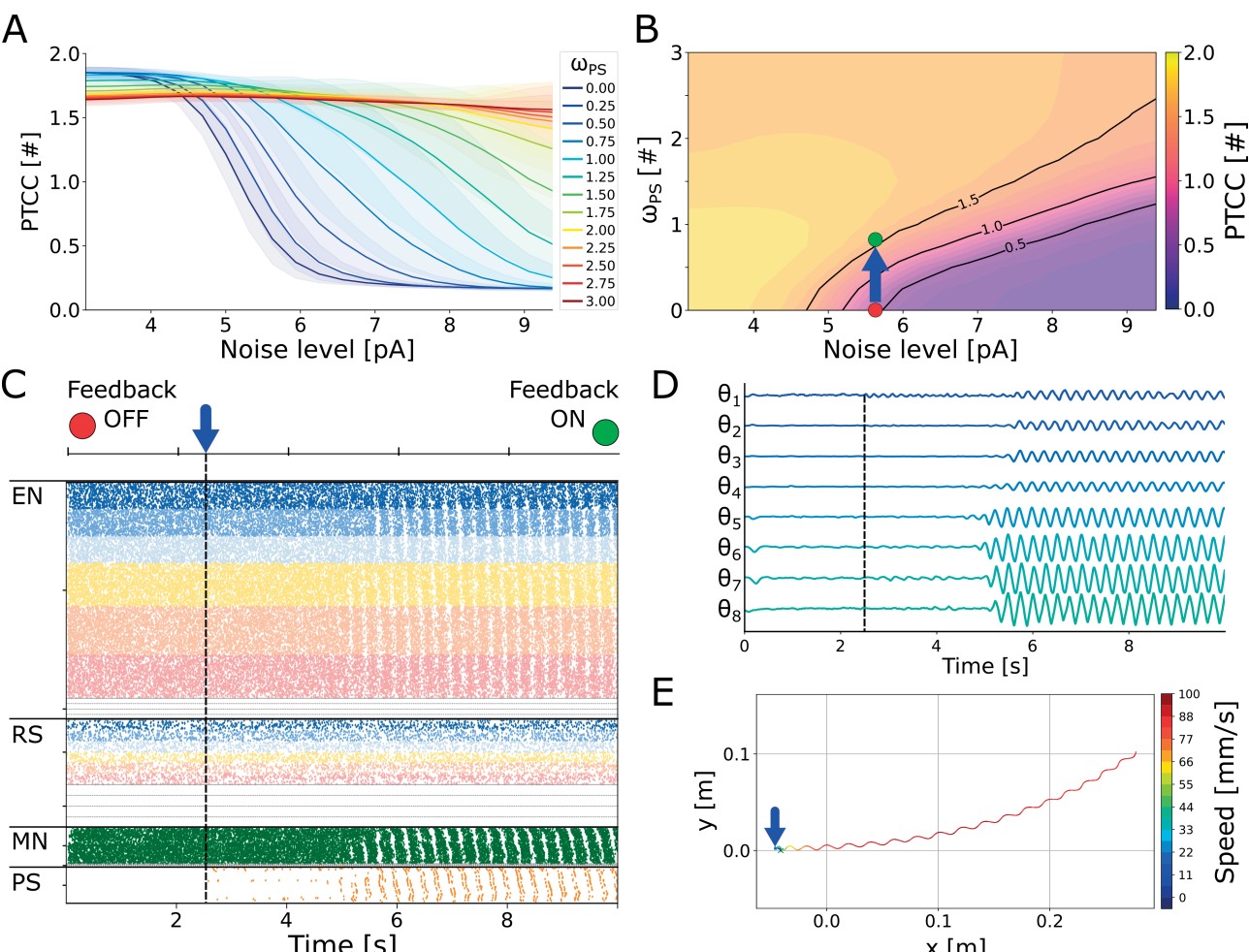

**Fig 8. Effect of noise on the closed loop swimming network.** **A**) shows the effect of noise amplitude on the rhythmicity of the network (PTCC) with different levels of sensory feedback weight ($\omega_{PS}$). The noise level was varied in the range [3.0, 9.5] pA. The feedback weight was varied in the range [0.0, 3.0]. The rheobase angle ratio was set to $\theta_{RH}^{\%} = 10\%$. The different curves represent the average PTCC values across 20 instances of the network (built with different seeds for the random number generator) for different levels of $\omega_{PS}$ (shown in the legend). The shaded areas represent one standard deviation of difference from the mean. **B**) displays the noise-feedback relationship in 2D. For a noise level of 5.6 pA, increasing $\omega_{PS}$ from 0 (red dot) to 0.7 (green point) leads to a switch from irregular to organized rhythmic activity. **C**) shows the switch effect with a noise level of 5.6 pA. The raster plot represents the spike times of excitatory neurons (EN), reticulospinal neurons (RS), motoneurons (MN) and proprioceptive sensory neurons (PS). Only the activity of the left side of the network is displayed. At time = 2.5 s (blue arrow), the sensory feedback weight is changed from $\omega_{PS} = 0$ (Feedback OFF) to $\omega_{PS} = 0.7$ (Feedback ON), restoring a left-right alternating rhythmic network activity. **D**) shows the axial joint angles during the simulation in C. Each angle is shown in the range [−25°, +25°]. **E**) shows the center of mass trajectory during the simulation in C. Activating feedback restored forward swimming.

the threshold for the activation of the PS neurons (i.e., the rheobase angle $\theta_{RH}$, see Eq 14). Indeed, when $\theta_{RH}^{\%}$ was set to 30%, the expansion in the noise rejection region was markedly lower (see Fig L in S5 File).

Overall, sensory feedback and fluid-body dynamics were shown to improve the robustness of the swimming locomotor networks. Future biological experiments could explore this aspect using semi-intact preparations where dorsal roots are kept intact and the caudal joints are allowed to move, similarly to [37,109,110]. These experiments could add noise to the locomotor circuits by means of external current injection and evaluate the different responses of

in-vitro and semi-intact preparations. This investigation could offer a more comprehensive understanding of the interaction between sensory feedback and rhythmic locomotor activity in salamanders.

**Sensory feedback expands the rhythm-generating capabilities of the network.** In section "Open loop sensitivity analysis" in S5 File, it was shown that the open-loop network is highly sensitive to a modulation of the commissural inhibition weight (see Fig Ciii in S5 File). Similarly, in section "Closed loop sensitivity analysis" in S5 File, the analysis was repeated for different values of sensory feedback strength (see Fig F in S5 File). Interestingly, the analysis showed that sensory feedback can reduce the sensitivity of the network to the internal parameters, reducing the dependency of the system on a precise selection of the neural and synaptic parameters. Nonetheless, the weight of the commissural inhibition remained the parameter associated to the highest sensitivity indices for all the considered metrics. For this reason, we investigated its effect on the closed loop network, as well as its interplay with the stretch sensory feedback (see question 4). We examined the effect of varying inhibition strength with different $\omega_{PS}$ values (see Fig 9). The commissural inhibition strength was varied in the range $[0.0, 6.0]$. The feedback weight was varied in the range $[0.0, 5.0]$. For each combination of the parameters, 20 instances of the network were built with different seeds for the random number generation and the results are reported in terms of mean and standard deviation of the computed metrics. For an equivalent analysis in absence of sensory feedback, see subsection "Effect of commissural inhibition" and Fig D in S5 File.

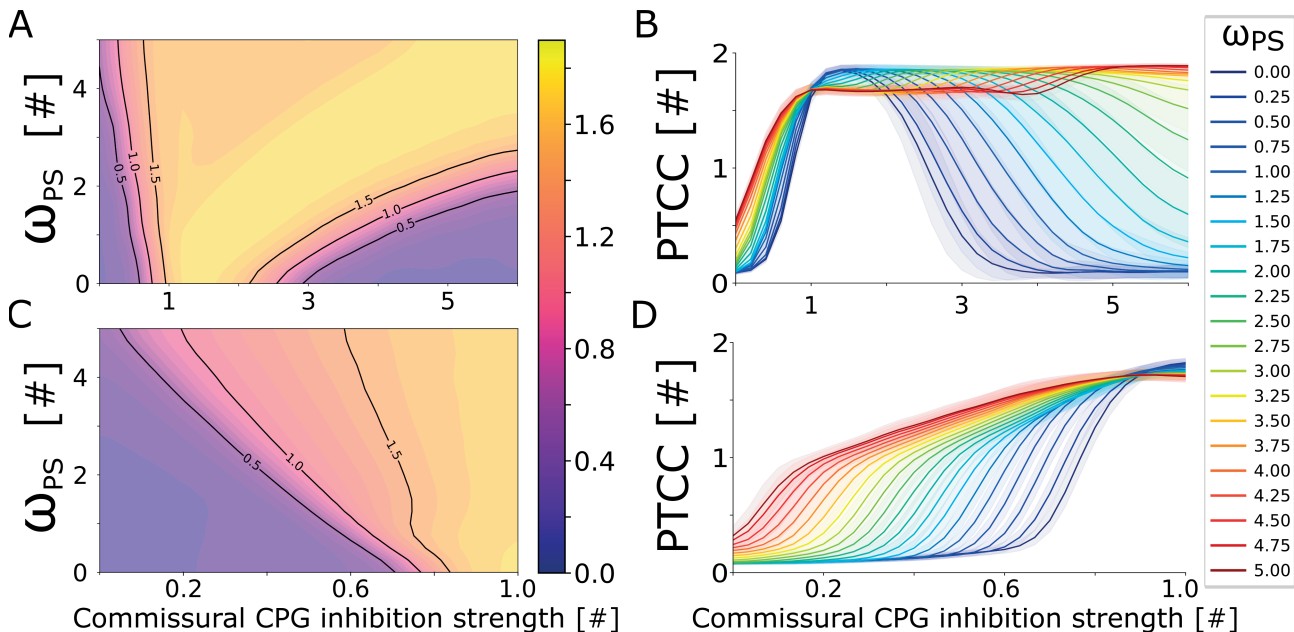

**Fig 9. Sensory feedback extends the rhythmogenic capabilities of the network. A)** shows the dependency of the periodicity (PTCC) on the inhibition strength and on the feedback weight ($\omega_{PS}$) for the swimming network. The commissural inhibition strength was varied in the range $[0.0, 6.0]$. The feedback weight was varied in the range $[0.0, 5.0]$. The rheobase angle ratio was set to $\theta_{RH}^{\%} = 10\%$. The displayed values are averaged across 20 instances of the network, built with different seeds for the random number generator. **B)** shows the inhibition-feedback relationship in 1D. The different curves represent the average PTCC values for different levels of $\omega_{PS}$ (shown in the legend). The shaded areas represent one standard deviation of difference from the mean. Panels **C** and **D** follow the same organization as the previous panels to highlight the inhibition-feedback relationship for $\omega_{PS}$ values in the range $[0.0, 1.0]$.

It was observed that as $\omega_{PS}$ increased, the region of stable oscillations expanded, both at low and high inhibition strengths. At higher inhibition levels, sensory feedback delayed the onset of the 'winner-takes-all' configuration (i.e.; one side becoming tonically active while the other remains silent, see right panel of Fig Dii in S5 File) by providing an extra burst termination mechanism for the active side, allowing the opposite side to escape ongoing inhibition (see Fig 9A and 9B). This is consistent with observations from models of the lamprey locomotor networks [1,43]. Interestingly, for high values of $\omega_{PS}$ the 'winner-takes-all' configuration completely disappeared and the network displayed oscillations for the entire range of studied commissural inhibition strengths.

At lower inhibition strength levels, sensory feedback helped channel the unpatterned oscillations of the open-loop network, reinstating an ordered left-right alternation pattern (see Fig 9C and 9D). Interestingly, this is in accordance with previous modeling studies on *C. Elegans* where coordinated swimming activity could be achieved in absence of central pattern generation from the axial circuits [111,112]. Overall these two effects lead to an expansion of the stable oscillation region by 63% for $\omega_{PS} = 1$. For very high feedback values (i.e., $\omega_{PS} > 2$), the region of stable oscillations was wider than the considered interval of inhibition strengths.

Interestingly, when the rheobase angle ratio $\theta_{RH}^{\%}$ (see Eq 14) was changed to 50%, the network's periodicity displayed a similar dependency on the inhibition strength and feedback weight (see Fig M in S5 File). In particular, the 'winner-takes-all' configuration was again deleted for high $\omega_{PS}$ values thanks to the feedback-mediated burst termination mechanism (see Fig Mi and Mii). On the other hand, for low levels of commissural inhibition strength, the effect was sensibly weaker (see Fig Miii and Miv). This is due to the incapability of the network to produce bending angles capable of reaching the higher activation threshold (i.e.; $\theta_{RH}$) when both sides of the spinal cord are tonically active. Nonetheless, the stable oscillation region was expanded by 16% for $\omega_{PS} = 1$.

The findings highlight the duality in rhythm generation mechanisms, where both central pattern generation and sensory feedback play essential roles [113]. Particularly, the model shows how intermediate levels of sensory feedback not only preserve but enhance the network's capability to generate rhythmic locomotor patterns, emphasizing the need for a balanced integration of central and peripheral inputs for optimal locomotor performance (see question 4).

Future biological experiments could further investigate the results of the current work using semi-intact preparations, where the caudal part of the body is left free to move and the strength of commissural inhibition is adjusted either pharmacologically or optogenetically. Through such experimental setups, we could gain a deeper understanding of the modulatory impact of sensory feedback on rhythmogenic network properties under different inhibition strengths.

## Limitations of the model

The current network architecture serves as a simplified model for the locomotor circuits of the salamanders. The biological motor circuits likely express different classes of interneurons (e.g., V2a, V0d, V0v [1,12]) that were not explicitly modeled in this network iteration. The advancements in genetic identification of the neuronal types within the salamander spinal cord will allow to refine the number and identity of neuronal classes of the simulated network. Additionally, this identification will allow to gather class-specific electrophysiological data that will help characterizing the neuronal and synaptic properties of the network [1].

Interestingly, [114] showed that the activity of the population in correspondence with the lumbar spinal cord of the turtle covers all phases, rather than oscillating between two alternating states. The recording area covered three spinal segments (D8, D9, D10), comprising the CPG oscillators for all limb degrees of freedom (hip, knee, ankle), the corresponding axial oscillators, as well as the limb and axial sensory neurons. If we were to mimic the same experiment in our model, the recording area would include the pre-girdle axial neurons, post-girdle axial neurons, and the limb oscillator neurons with various delays for the different degrees of freedom of the limbs. Consequently, despite its simplicity, our model would also almost uniformly cover all the neural phases. Furthermore, including additional types of interneurons and sensory neurons would likely further contribute to spreading the phases covered by neurons during locomotor rhythms, aligning with the results in [114].

With respect to the limbs network, the current model only accounted for a single equivalent segment to simulate the rhythmic limb activity during locomotion, using delayed versions of the same signal to control the different degrees of freedom. On the other hand, several studies revealed that distinct pools of interneurons govern the activity of each joint within the limbs [115,116]. Future iterations of the network should account for this limitation in order to investigate the emergence of intra-limb coordination in salamanders. Additionally, the inter-limb connectivity of the current model was designed to generate coordination between diagonally opposed limbs, as observed during trotting gaits [61]. Different inter-limb connectivity schemes could allow to simulate a richer variety of patterns (e.g., lateral and diagonal sequence walks) expressed by salamanders during ground locomotion [72].

Given the limitations of the walking network in the current model, this study focused on the effect of axial proprioceptive sensory feedback during swimming. Future studies will evaluate whether the beneficial effects of sensory feedback identified in the current work still exist during walking. More generally, the effect of the body dynamics (e.g. body stiffness and damping) and the environment dynamics (change of viscosity, etc.) should also be investigated to get a deeper understanding of the interplay between central pattern generation, mechanical properties and sensory feedback [111,117,118].

Finally, the proposed model could also be used to investigate the effects of other sensory modalities on the locomotor circuits of the salamander. In particular, the effect of exteroceptive sensory feedback from the hydrodynamic forces acting on the body during swimming could be analysed to extend the results from previous robotics studies [52]. Similarly, the walking network could receive pressure feedback from the ground reaction forces acting on the body and the limbs. Interestingly, the sensory feedback from ground interaction forces might explain the generation of S-shaped standing waves during walking [119]. Additionally, this sensory modality would allow to investigate the emergence of interlimb coordination through sensory feedback [120,121].

## Conclusions

In this study, we have developed a novel neuromechanical model to investigate the interplay between CPGs and proprioceptive feedback in governing the locomotor behaviors of salamanders. Our findings underline the intricate balance between central control and sensory feedback in producing adaptable and robust locomotor patterns in a bio-inspired network architecture.

The proposed model replicates several findings from fictive locomotion experiments in salamanders and it was shown to be able to generate oscillations at the level of full cord, full hemicord, isolated segments and isolated hemisegments [93,94]. When including the

descending drive from reticulospinal neurons, our simulations demonstrate that the rhythmic locomotor outputs, essential for swimming and walking, can be dynamically modulated by varying the excitatory drive to the network, underscoring the importance of central neuronal mechanisms. The model effectively replicates the physiological propagation of motor patterns seen in natural salamander locomotion, highlighting the role of CPGs in generating basic motor rhythms [1,12,93]. Interestingly, this is the first spiking neuromechanical model of the salamander locomotor networks that can display both swimming and walking with a unified newtwork architecture (see question 1). Additionally, the open-loop network responds to increasing levels of external stimulation similarly to previous simulation studies, suggesting that the model assumptions are well justified through a chain of controllers, ranging from detailed bio-physical networks (e.g., [81]) through formal spiking models (e.g., [89] and present model), all the way to more abstract controllers [69,72].

Incorporating sensory feedback into our model revealed its pivotal role in refining locomotor efficiency and adaptability [12]. Indeed, intermediate levels of sensory feedback lead to an increase of the speed and a reduction of the energetic cost of transport during swimming compared to the open loop network (see question 2). Additionally, the feedback mechanisms, modeled on different topologies observed in nature [42–44] showed similar trends in the modulation of frequency and phase lag, irrespective of the type (excitatory, inhibitory) or directionality (ascending, descending) of the feedback projections (see question 3).

When dealing with badly tuned CPG networks (e.g., high/low drive, high/low commissural inhibition), the feedback enhanced the network's robustness by effectively extending the functional range of the system [52,122] (see question 4). This suggests a broader role of sensory feedback in locomotor control than previously appreciated, extending beyond mere fine-tuning to actively shaping the motor output and allowing it to work in a larger range of conditions than open-loop networks.

Our analyses indicate that sensory feedback remarkably contributes to the locomotor network's robustness, enabling it to withstand high levels of noise and variability in neuronal parameters [52] (see question 4 and question 5). For intermediate values of sensory feedback strength, this robustness is achieved without compromising the flexibility of the system. Indeed, in this case the network retains the ability to adjust its locomotor patterns in response to changes in external drive and sensory input, echoing the adaptive nature of biological systems. Conversely, high values of feedback weight lead to uncontrollable networks where frequency and phase lags could not be modulated through descending commands (see question 2).

Our findings have significant implications for the design of bio-inspired robotic systems, suggesting that incorporating models of proprioceptive feedback can substantially improve the adaptability and robustness of locomotor control in bio-inspired robots [52,72,82]. Furthermore, the insights gained from this study contribute to our understanding of the evolutionary adaptations in vertebrate locomotion, offering a computational perspective on how sensory feedback mechanisms might have evolved to complement central control strategies across different species.

This work opens avenues for future research to explore the effects of more complex sensory feedback mechanisms, including those involving other modalities such as tactile and visual inputs [1,123]. Notably, the current paper mainly focused on the analysis of the swimming locomotor networks. In the future, the investigation should be expanded to include the walking network and the effect that proprioceptive (e.g., shoulder and hip flexion/extension) and exteroceptive (e.g., ground reaction forces acting on the limbs) feedback modalities have on the resulting locomotor activity. Additionally, experimental validation of the model's predictions in biological systems, particularly through manipulation of sensory feedback

pathways, could provide deeper insights into the neuromechanical underpinnings of locomotion [12,44]. Finally, the model could be used to investigate the possible role of sensory feedback during regeneration after a spinal cord injury, as recently proposed in [51] for the lamprey.

In conclusion, our model advances the understanding of the multifaceted control mechanisms underlying salamander locomotion, bridging the gap between simplistic representations and the complex realities of biological locomotion. By highlighting the synergistic roles of CPGs and proprioceptive feedback, this work lays the groundwork for future explorations into the neuronal basis of movement in vertebrates, with potential applications in robotics, rehabilitation, and the study of locomotor evolution.

## Methods

### Neuronal model

In this model, neurons are modeled as adaptive integrate-and-fire (hereafter referred to as IF) neurons [124], which are essentially formal spiking neurons, augmented with an adaptation variable as

$$\tau_{memb}\frac{du}{dt} = -(u - E_{rest}) - R_{memb}\omega + R_{memb}I_{syn} + R_{memb}I_{ext}(t) \tag{1}$$

$$\tau_\omega\frac{d\omega}{dt} = -\omega \tag{2}$$

In this context, $u$ represents the membrane potential, $E_{rest}$ signifies the neuronal resting potential, $R_{memb}$ is the input membrane resistance, $I_{syn}$ represents the total synaptic current, $I_{ext}$ is the external input current, and $\omega$ denotes the adaptation variable. The neuronal model integrates inputs and, upon reaching the firing threshold $E_{th}$, the spike time of a neuron is documented followed by a reset of the membrane potential to the reset voltage $E_{rest}$. Each spike occurrence increments the adaptation variables by a value of $\Delta\omega$. Moreover, post-spike, the neuron is held at the resting membrane potential for a neuron type-dependent refractory duration $t_{refr}$.

The total synaptic current is given by the contribution of different synaptic conductances $g_{syn}$ and their respective synaptic reverse potential $E_{syn}$ according to

$$I_{syn} = \sum_{syn} g_{syn}(u - E_{syn}) \tag{3}$$

The conductance $g_{syn}$ for both inhibitory and excitatory synaptic currents sees an increment by $\Delta g_{syn}$ with every incoming spike [125]. Additionally, there is a constant synaptic transmission delay of 2ms between the spike event and the corresponding $g_{syn}$ modification. The temporal evolution of synaptic conductances is dictated by

$$\tau_{syn}\frac{dg_{syn}}{dt} = -g_{syn} \tag{4}$$

The excitatory postsynaptic potentials (EPSPs) embody a swift *alpha*-amino-3-hydroxy-5-methylisoxazole-4-propionic acid (AMPA)-like component alongside a slowly diminishing N-methyl-D-aspartic acid (NMDA)-like component, taking cues from lamprey findings [126] and salamander modeling endeavors [81,89]. Conversely, the inhibitory postsynaptic potentials (IPSPs) act on a single glycinergic (GLYC)-like component.

When noise was included in the network, it was implemented as an Ornstein-Uhlenbeck random process [127,128] added to the membrane voltage of the neurons as a current

$$I_{noise} = \sqrt{\frac{2}{\tau_{memb}}} \xi(\sigma_{noise}, t) \tag{5}$$

where $\xi(\sigma_{noise}, t)$ is a zero-mean Gaussian white noise process with standard deviation $\sigma_{noise}$ denoting the strength of the disturbance.

**Muscle cells model.** Muscle cells (MC) represent an additional non-spiking cell population added to the model in order to transform the discrete spiking trains coming from the simulated neurons into a continuous output signal which is then used to compute the output torque. They are described by

$$\tau_{mc} \frac{dM_{side_i}}{dt} = -M_{side_i} + RI_{syn_{mc}} \tag{6}$$

$$I_{syn_{mc}} = \sum_{syn_{mc}} g_{syn_{mc}} (M_{side_i} - E_{syn_{mc}}) \tag{7}$$

Where, similarly to Eq 1, $\tau_{mc}$ is the membrane time constant, $M_{side_i}$ is the membrane potential ($side \in L, R$), $I_{syn_{mc}}$ is the total synaptic current, $g_{syn_{mc}}$ is the synaptic conductance and $E_{syn_{mc}}$ is the synaptic reverse potential. Effectively, the muscle cells are modeled as firing rate neurons [84] that act as low-pass filters of the input activity. Their input is normalized in the range $[0.0, 1.0]$ and can be interpreted as the degree of activation of the motor neurons in the corresponding network region.

## Network model

The salamander network comprises 5 different classes of neurons, namely excitatory neurons (EN), inhibitory neurons (IN), motoneurons (MN), reticulospinal neurons (RS) and propriosensory neurons (PS). The populations differ for their neuronal and synaptic properties, as well as for their connectivity patterns with other neurons. The values of the neuronal and synaptic properties of the network are reported in S2 File.

Figure 1A depicts conceptual schematics for the network model of the salamander spinal circuitry. Neurons of the axial network are arranged uniformly in a continuous two-dimensional sheet without explicit segmental boundaries. The network is 10cm long, in accordance with the typical length of *Pleurodeles waltl*'s spinal cord [129]. Each neuron is assigned a coordinate ($x_{neur}$) specifying its position along the spine starting from the rostral part. Additionally, the axial network is divided in two hemicords built according to the same organization principles. For this reason, each neuron is also assigned a second coordinate ($y_{neur}$) indicating its position on the left or right hemicord. The coordinate system introduced for the network organization allows for the definition of position-dependent connections between neurons in order to target specific regions of the spinal cord as well as the most proximal neurons for each candidate. Connection densities and weights in the salamander spinal cord are largely unknown and have been treated as open parameters.

Since the salamander spinal cord exhibits 40 vertebrae and ventral roots, equally spaced consecutive sections of the network can be viewed to represent a spinal segment (henceforth termed an equivalent segment), with a length of 2.5mm. Note that the notion of equivalent segment is only used to quantify the local activity of a region along the continuous axial network (see Fig Ai and Aiii in S1 File) but it does not influence the way the network is built.

Indeed, all connections are defined based on probability distributions taking into account solely the distance, side and type of the affected neurons.

Each equivalent segment accounts for 60 EN and 60 IN, representing the CPG component of the network (see Fig 1C). The number of CPG neurons was derived by [89,116]. Additionally, each equivalent segment comprises 10 MN and 10 PS units. The number of motoneurons was inspired from [130,131]. The number of proprioceptive sensory neurons was inspired by studies in the lamprey and zebrafish [44,45]. Finally, each equivalent segment includes 2 muscle cell units (MC) whose role is to integrate the spiking activity of the motoneurons. Note that the classes and numbers of neurons in the salamander locomotor networks is largely unknown and it could only be estimated in the present work. The model could be further refined with future advancements in imaging and genetic identification of the salamander spinal interneurons [12].

Fig 1C depicts a schematic of the neuronal organization of each equivalent segment. We follow the convention of denoting each segment by two oscillator symbols since equivalent hemisegments can independently generate rhythmic activity (see section "Simulating fictive locomotion" and Fig Aiv in S5 File). This result is a consequence of the recurrent excitation within hemisegments, coupled with the adaptive nature of the CPG neurons. Notably, including pacemaker currents was not necessary to generate rhythm at the level of isolated spinal segments in the model (see Fig Aiii in S5 File). Interestingly, in-vitro experimental results showed that abolishing the persistent sodium current via bath application of riluzole in surgically-isolated spinal segments did not affect the frequency and rhythmicity of the corresponding ventral root oscillations [93]. This result indicates that pacemaker currents may not be necessary for rhythm generation at a segmental level in salamanders.

Separate RS populations of 100 excitatory neurons each project to the rostral portion (Rost), the mid-trunk (Midt), the end of the mid-trunk (Endt), the pelvic girdle area (Pelv), the tail (Tail), and the caudal end (Caud) regions of the axial network, with distinct connectivity patterns (cf. Fig 1B, dashed lines indicating incoming descending connection density from RS nuclei). The connection probabilities of the RS populations are given by overlapping trapezoidal distributions. The parameters of the distributions are listed in Table E in S2 File. The overlapping of the RS projections is reduced between the end of the mid-trunk and the pelvic girdle area (see Fig 1B). This reduced connectivity permits to effectively decouple the control of the trunk and tail regions, thus allowing the two parts to be independently activated and modulated, consistently with experimental findings [59,97,99].

The connectivity between neurons within the axial network is obtained by means of asymmetric exponential distributions centered in correspondence with the pre-synaptic neuron. The probability distribution as a function of the pre-synaptic ($x_{pre}$) and post-synaptic ($x_{post}$) neurons positions is defined as

$$P(x_{pre}, x_{post}) = \begin{cases} Ae^{-\frac{(x_{pre}-x_{post})^2}{4\sigma_{up}^2}}, & \text{if } x_{pre} >= x_{post} \\ Ae^{-\frac{(x_{pre}-x_{post})^2}{4\sigma_{dw}^2}}, & \text{if } x_{pre} < x_{post} \end{cases} \tag{8}$$

Where $A$ specifies the maximum connection probability, while the parameters $\sigma_{up}$ and $\sigma_{dw}$ determine how wide the probability distributions are for ascending (i.e. rostrally oriented) and descending (i.e. caudally oriented) connections respectively. The parameters of the connections for the axial network are listed in Table F in S2 File.

Within the axial CPG network, the connections are biased towards more caudally located neurons, as suggested by physiological and modeling studies in the lamprey [4,80,132,133] (see 1C). Conversely, the axial MN populations project symmetrically to the closest neighbouring muscle cells, thus providing an estimate of the activity of the local axial region. In the axial network, the PS neurons target the closest EN, IN and MN neurons both ipsilaterally and contralaterally. Previous experimental results from lamprey and zebrafish highlighted an asymmetry of the PS projections in favour of dominantly ascending connections [41,44,45]. For this reason, the effect of different PS topologies (excitation or inhibition, ascending or descending projections) was also studied and it is discussed in this work (see sections "The effect of axial sensory feedback is largely consistent across different feedback topologies" and "The neuromechanical principles of frequency and phase-lag control by stretch feedback during swimming").

Similarly to the RS connectivity, the range of the connections between the end of the mid-trunk and the pelvic girdle area is reduced to 0.75mm (i.e. one third of an equivalent segment). The discontinuity allows the trunk and tail sub-networks to independently generate waves of activation that do not propagate to the other section when not excited from its corresponding RS modules (more details in the next sections), in accordance to experimental findings [97,99].

In addition to the axial network, four sub networks are modeled to simulate the activities of the limbs. Two limb pairs are placed in correspondence of the first axial segment ($X = 0.0$ mm) and of the first pelvis segment ($X = 37.5$ mm) respectively (see Fig 1B, green circles). Separate oscillatory networks for each limb are implemented as equivalent segments, analogously to the axial network. Note that this is a simplification that does not take into account the possible presence of separate oscillatory pools for each distinct degree of freedom [115, 116]. Also, the current model does not take into account the presence of a pattern formation layer, which is generally accepted in other animal models such as cats and mice [134–137].

The two half centers of each limb oscillator represent the flexor and extensor components, respectively. The limbs project to the pelvic and scapular region of the axis through local exponentially decaying probabilities (see Eq 8 ) with $A = 0.1$, $\sigma_{up} = 0.0$ mm, $\sigma_{dw} = 5.0$ mm (see Fig 1B). The connections involve projections from the flexor EN to the ipsilateral axial network, as well as projections from the flexor IN to the contralateral axial network (see Fig 1D). The synaptic connections between the four limb CPGs were designed to resemble the activation patterns displayed by salamanders during trotting gait [92] (see Fig 1E). The inter-limb projections were drawn between the IN populations of the flexor side of the limbs segments. The connections targeted ipsilateral limbs (i.e., limbs on different girdles located on the same side of the body) and commissural limbs (i.e., limbs on the same girdle located on opposite sides of the body). The two connection types ensured a synchronous activation of the diagonally opposed limbs segments. The density of inter-limb connections was lower than the one within the limb hemisegments in order to avoid the disruption of their rhythm generation properties. In animals, the inter-limb connectivity may be subject to modulation depending on the expressed gait pattern [39,101,102]. A full analysis of the possible inter-limb connectivity schemes and their implication in inter-limb coordination is outside the scope of the current work.

The synaptic weights were adjusted manually by modulating the corresponding $\Delta g_{syn}$ parameter (see Eq 4) and are listed in Table C in S2 File. The weights for the PS populations are additionally modulated by an adimensional gain $\omega_{PS}$ (see Fig 2E and 2F) according to

$$\Delta g_{syn}^* = \Delta g_{syn} \omega_{PS} \qquad (9)$$

Where $syn \in \{ampa, nmda, glyc\}$ and $\Delta g_{syn}$ is the nominal value listed for the PS populations in Table C in S2 File. The role of $\omega_{PS}$ is to represent the relative strength between PS connections and intra-CPG connections. When $\omega_{PS}$ is set to 1, the PS-originating synapses have the same strength as the connections between EN and IN populations within the locomotor circuits. More details about the role of $\omega_{PS}$ are discussed in the next sections.

## Mechanical model

The mechanical model consists of a salamander body, inclusive of axial and limb joints, that was previously introduced in [91]. The axial configuration is a chain of 9 links, interconnected by 8 active joints (i.e., driven by muscle models) functioning in the horizontal plane (see Fig 2B, red arrows). Moreover, the axial structure incorporates 6 passive joints positioned at the mid trunk (2 joints) and tail (4 joints) sections (see Fig 2B, blue arrows). These passive joints are activated solely by the passive elements of the muscle model, they operate in the sagittal plane and they enable the belly and tail to engage with the ground. The model also comprises four limbs, with each limb accounting for 4 degrees of freedom (DoF). These encompass flexion/extension, abduction/adduction, and internal/external rotation of the shoulder (or hip) along with flexion/extension of the elbow (or knee, see Fig 2B, bottom right insert). The geometrical and dynamical properties of the model were derived from *Pleurodeles waltl* biological data. More details about its design are given in [91]. For simplicity, the density of the model was considered constant across the length of the body and equal to $981.0 kg/m^3$. This value renders the model positively buoyant in water, thus allowing to study swimming behaviors during surface-level aquatic locomotion. In the future, further refinements of the model could aim at varying the density of the segments based on the differential amount of muscles, bones and other tissues. The cross-section of the model varies with its axial position, resulting in a heavier head and trunk section compared to the lightweight tail section.

## Electro-mechanical transduction

The activity of muscle cells is transduced in output torques for the mechanical model using the muscle model proposed by Ekeberg [138] (see Fig 2C). For each joint $i$, the model receives in input the flexor ($M_{L_i}$) and extensor ($M_{R_i}$) activations from the muscle cells (see Eq 6), and the current joint angle ($\theta_i$) and speed ($\dot{\theta}_i$) to compute the resulting output torque ($\tau_i$) via:

$$\tau_i = \alpha_i M_{diff_i} + \beta_i(\gamma_i + M_{sum_i})(\theta_i - \theta_{0_i}) + \delta_i \dot{\theta}_i \tag{10}$$

$$M_{diff_i} = G_{MC_i}(M_{L_i} - M_{R_i}) \tag{11}$$

$$M_{sum_i} = G_{MC_i}(M_{L_i} + M_{R_i}) \tag{12}$$

Where we $\alpha_i$ is the active gain, $\beta_i \gamma_i$ is the passive stiffness, $\beta_i M_{sum_i}$ is the active stiffness, $\delta_i$ is the damping coefficient, $\theta_{0_i}$ is the resting angle position. The flexor and extensor activations are scaled by a factor $G_{MC_i}$ that is varied across different joints based on the salamander kinematics (more details in the following paragraphs). The full electro-mechanical pipeline is displayed in Fig 2F. The segmental activity information from the axial CPG flows through the motoneurons and it is temporally and spatially integrated by the muscle cells. The muscle cells consequently provide the activation signal for the Ekeberg muscles that, in turn, output the total torque generated by the corresponding joint. Effectively, the muscle model is a spring-mass-damper system with variable stiffness which results in a non-linear input-output relationship (see Fig 2C). This implementation is a simplified version of the widely used Hill-type

muscle model [139] that, while being more numerically tractable, still comprises active and passive components of muscle activations.

In the axial joints, the muscle parameters were chosen in order to operate in a target dynamical regime in terms of resonance frequency, damping factor and zero-frequency gain of the Ekeberg muscles. This selection allows to scale the muscle properties of the different joints so that they match the different inertial forces they are subjected to during locomotion. A detailed description of the muscle parameters optimization is reported in S3 File.

During swimming, the resting angles of the limb joints were set in order to fold them along the body axis [92]. During walking, since a single equivalent segment was used for each limb, delayed copies of the same muscle cell (MC) signals were used to control its multiple degrees of freedom to provide the periodic limb movements necessary for ground locomotion. In order to achieve that, the period of each limb's oscillations was estimated online with a threshold-crossing strategy. The threshold for the membrane potential of the muscle cells was set to the same value for all the limbs ($M_{side_i}$ = 0.25, see Eq 6). The times of threshold crossing were registered and the corresponding limb period was estimated as the difference between two successive threshold times. The different degrees of freedom were then driven by a delayed version of the same MC signals. Similarly to [73], the delays chosen for the DoF were applied as a fraction of the estimated period. The delay fraction was set to 0% for the shoulder abduction/adduction degree of freedom, which thus acted as the leading joint. Conversely, the delay was set to 25% for the shoulder flexion/extension, shoulder internal/external rotation and elbow flexion/extension to ensure ground clearance during the swing phase.

## Mechano-electrical transduction

The proprioceptive feedback neurons residing in the stretched side of the body receive an input that is proportional to the angles of the mechanical joints according to:

$$I_{PS} = max(0, G_{PS}^i \cdot \theta * i) \tag{13}$$

Where $G_{PS}^i$ is the gain for the mechano-electrical transduction for the i-th segment (see Fig 2D and 2E). The angle input $\theta^i$ provided to each sensory neuron is obtained from the linear interpolation between the two closest mechanical joints, thus assuming a continuous curvature gradient along the body. The value of the sensory feedback gain for each segment was chosen based on the rheobase current of the sensory neurons ($I_{RH}$ = 50$pA$) so that the critical angle at which the neurons start to fire (hereafter denoted rheobase angle $\theta_{RH}^i$) scales proportionally with respect to the reference amplitude $\Theta^i$ of the corresponding joint oscillations, derived from [61]. The relationship is expressed as:

$$\theta_{RH}^{\%} == \frac{I_{RH}}{G_{PS}^i \Theta^i} \cdot 100\% = \frac{\theta_{RH}^i}{\Theta^i} \cdot 100\% \tag{14}$$

Where $\theta_{RH}^{\%}$ denotes the rheobase angle ratio (expressed as a percentage of $\Theta_i$), and it is equal for all the segments of the network (see Fig 2D). This was done according to the hypothesis that the stiffer body joints will be more sensitive to stretch with respect to the flexible tail joints. The scaling also ensures that the activation of the sensory neurons happens in the same phase of the oscillation regardless of their amplitude. Indeed, the rheobase angle ratio effectively controls the percentage of a joint oscillation period for which the sensory neurons are active, thus modulating the duration of the feedback action. Unless stated otherwise, the rheobase angle ratio for each segment was set to 10%. Several analyses were also carried out with different $\theta_{RH}^{\%}$ values (e.g., 30% and 50%) and the corresponding figures can be found in

S5 File. Notably, the results for $\theta^\%_{RH}$ = 30% typically showed the same trends as the ones for $\theta^\%_{RH}$ values of 10% and 50%. For this reason, they were omitted in favor of showing only the extreme cases for $\theta^\%_{RH}$ = 10% (in the main text) and $\theta^\%_{RH}$ = 50% (in S5 File). Future biological studies could investigate whether salamanders present a gradient in the degree of excitability of the sensory neurons along the spine and, consequently, what is the critical spine curvature that leads to their activation.

Overall, the effect of sensory feedback is mainly modulated by two parameters, namely the $G_{PS}$ gain and $\omega_{PS}$ (see Fig 2D and 2E). The former affects the rheobase angle ratio $\theta^\%_{RH}$. The latter modulates the influence that an activation of PS has on the other neurons (see Eq 9). In an analogy with proportional control, the two parameters modulate the threshold value ($\theta_{RH}$) and slope ($W$) of the overall sensory signal provided to the locomotor network (see Fig 2E).

## Hydrodynamic model

When submerged in water, the body is subjected to buoyancy and hydrodynamic forces. Buoyancy forces are applied to each link according to its submerged volume $V_{UW}$

$$\mathrm{F}_{b_i} = V_{UW_i}\rho g \tag{15}$$

Where $\rho$ is the water density and $g$ is the gravity acceleration ($g = 9.81 m/s^2$). Since salamanders typically swim in high Reynolds number regimes, we opted for a simplified hydrodynamic model composed of inertial drag forces, analogously to [138,140]. For each link, a speed dependent drag force ($F_d$) is applied in each dimension ($i$) according to:

$$\mathrm{F}_{d_i} = \frac{1}{2}\rho C_{d_i} A_i V_i^2 \tag{16}$$

Where $A$ is the link's cross-sectional area, $c_{d_i}$ is the drag coefficient in the i-th dimension, $V_i$ is the current speed of the link in the i-th dimension in the reference frame of the center of mass of the link. The drag coefficients were chosen in order to match the mechanical metrics computed from swimming salamanders. To achieve this, the muscle cells gains $G_{MC_i}$ (see Eq 10) were adjusted so that the angular excursions displayed by the joints would match the ones reported in [61]. Successively, the drag coefficients were hand-tuned to achieve a good level of agreement between the experimental and simulation metrics. A detailed description of the muscle cells factors selection and resulting swimming metrics is reported in S4 File. The selected coefficients allowed to closely match the experimental results in terms of axial displacements and speed (see Fig A in S4 File). An animation displaying the generated swimming activity can be seen in S1 Video.

## Neuromechanical simulation

The neural network was simulated with the Brian2 library [141,142] using the Python programming language. The tool allows to define custom neuronal and synaptic models, as well as to define connection probability distributions dependent on neurons' type, position and properties. Overall, the model is comprised of for 7'408 neurons and approximately 100'000 synaptic connections. Unless stated otherwise, the duration of each simulation was set to 10s. The network dynamics were integrated with forward Euler integration method and with a timestep of 1ms. Using Brian2 allows to convert the neural network into Cython code, boosting the simulation performance.

The mechanical model was simulated in synchrony with the neural model with the MuJoCo physics engine [143] via the FARMS simulation framework [91]. The tool provides a platform for the definition and simulation of neuromechanical simulations, as well as for the incorporation of hydrodynamic and muscle models.

The two simulations exchanged data via queue communication among their respective threads. The integration step of the mechanical model was set to 1ms (the same as the neural model) and data exchange was carried out at each timestep.

The data from the neuromechanical simulations were used to compute the corresponding neural and mechanical metrics to evaluate the performance of the model. The details about the signal processing pipeline can be found in S1 File.

### Sensitivity analysis

Sensitivity analysis is the study of how the uncertainty in the output of a system can be apportioned to different sources of uncertainty in its inputs. Studying the sensitivity of the network performance metrics (e.g., periodicity, frequency, phase lag) with respect to the network parameters helps identify the key variables influencing the system's behavior, offering insights into the importance of various design decisions. The sensitivity of each input is represented by a numeric value, called the sensitivity index. Total-order indices measure the contribution to the output variance caused by a model input, including higher-order interactions.

Simulations were performed using SALib [144,145], an open-source library written in Python for performing sensitivity analyses. The parameters were chosen using Saltelli's extension of the Sobol' sequence, a quasi-random low-discrepancy sequence used to generate uniform samples of parameter space [146,147]. The neuronal and synaptic parameters of the CPG network were varied for the analysis. In particular, the neuronal parameters included $\tau_{memb}$, $R_{memb}$, $\tau_\omega$, $\Delta\omega$ of the EN and IN populations. Similarly, the synaptic parameters included the weights $\Delta g_{ampa}$, $\Delta g_{nmda}$, $\Delta g_{glyc}$ of the EN and IN projections. Finally, when the sensitivity analysis was carried out in the full closed-loop network, the sensory feedback parameters $G_{PS}$ and $\omega_{PS}$ were also modified. During the analysis, each of the aforementioned parameters was sampled in a range between 0.95 and 1.05 times its nominal value.

Overall, each sensitivity analysis involved the simulation of 704 combinations of parameter values. Additionally, the analysis was repeated for 30 instances of the network, built with different seeds for the random number generator. For each iteration, the total-order sensitivity indices were computed with respect to each neural and mechanical metric.

### Supporting information

**S1 File. Signal processing pipeline.**
(PDF)

**S2 File. Neuronal and synaptic properties of the network.**
(PDF)

**S3 File. Optimization of the muscle parameters for the mechanical model.**
(PDF)

**S4 File. Matching of the salamander kinematics during anguilliform swimming.**
(PDF)

**S5 File. Supplementary results for the analyses described in the results.**
(PDF)

**S1 Video. Swimming salamander matching the experimental kinematics.**
(MP4)

**S2 Video. The generated swimming activity without sensory feedback.**
(MP4)

**S3 Video. The generated trotting activity without sensory feedback.**
(MP4)

**S4 Video. The generated swimming activity with sensory feedback.**
(MP4)

**S5 Video. Sensory feedback restoring the swimming activity in a noisy network.**
(MP4)

**S6 Video. Sensory feedback restoring the swimming activity in a very noisy network.**
(MP4)

## Author contributions

**Conceptualization:** Alessandro Pazzaglia, Andrej Bicanski, Andrea Ferrario, Dimitri Ryczko, Auke Ijspeert.

**Data curation:** Alessandro Pazzaglia, Andrej Bicanski.

**Formal analysis:** Alessandro Pazzaglia, Andrej Bicanski.

**Funding acquisition:** Dimitri Ryczko, Auke Ijspeert.

**Investigation:** Alessandro Pazzaglia, Andrej Bicanski.

**Methodology:** Alessandro Pazzaglia, Andrej Bicanski, Andrea Ferrario, Jonathan Arreguit, Dimitri Ryczko, Auke Ijspeert.

**Project administration:** Dimitri Ryczko, Auke Ijspeert.

**Resources:** Auke Ijspeert.

**Software:** Alessandro Pazzaglia, Jonathan Arreguit.

**Supervision:** Dimitri Ryczko, Auke Ijspeert.

**Validation:** Alessandro Pazzaglia.

**Visualization:** Alessandro Pazzaglia, Andrea Ferrario, Jonathan Arreguit.

**Writing – original draft:** Alessandro Pazzaglia, Andrea Ferrario.

**Writing – review & editing:** Alessandro Pazzaglia, Andrej Bicanski, Jonathan Arreguit, Dimitri Ryczko, Auke Ijspeert.

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
