## [Decision Letter · Decision Letter 0]

12 Aug 2024

Dear MSc Pazzaglia,

Thank you very much for submitting your manuscript "Sensory and central contributions to motor pattern generation in a spiking, neuro-mechanical model of the salamander spinal cord" for consideration at PLOS Computational Biology.

As with all papers reviewed by the journal, your manuscript was reviewed by members of the editorial board and by several independent reviewers. In light of the reviews (below this email), we would like to invite the resubmission of a significantly-revised version that takes into account the reviewers' comments.

The reviews of two reviewers are now completed. As you can see, there are major concerns raised by both reviewers, but specifically around the use of the model to replicate some basic motifs of the system. If you would like to resubmit the article, it is critical that you thoroughly address these review comments, which may include more simulations of motor control motifs, the integration of electrophysiology to demonstrate neuronal types, and clarity on the choice of neuron model.

We cannot make any decision about publication until we have seen the revised manuscript and your response to the reviewers' comments. Your revised manuscript is also likely to be sent to reviewers for further evaluation.

Sincerely,

David Borton

Guest Editor

PLOS Computational Biology

Thomas Serre

Section Editor

PLOS Computational Biology

The reviews of two reviewers are now completed. As you can see, there are major concerns raised by both reviewers, but specifically around the use of the model to replicate some basic motifs of the system. If you would like to resubmit the article, it is critical that you thoroughly address these review comments, which may include more simulations of motor control motifs, the integration of electrophysiology to demonstrate neuronal types, and clarity on the choice of neuron model.

Reviewer's Responses to Questions

**Comments to the Authors:**

Reviewer #1: The study by Pazzaglia A, etc. presents a spiking neural network dependent neuromechanical model and investigates the role of proprioceptive sensory feedback. There is novelty that the interplay of proprioceptive feedback and CPG has been studied in many other species, but rarely salamanders. There a few points may be helpful to present the work.

The proposed model fails to reproduce fundamental biological features of CPG, such as oscillation in segments.

The authors stated in the methods that the type and number of neurons are largely unknown and they are based their model on estimation, some from other kinds of animals. It is not impossible, or even difficult, to obtain such experimental data given current technologies. The results thus cannot lead to concrete conclusions of biological system. This may suit better as an electromechanical model for robotic control inspired by biology and simulation results showing features of such electromechanical model.

Is HH model or one fits all neural model enough to cover all different neuronal types in CPG? I am suspicious of this. But if yes, justification with experimental of literature evidences are needed.

The authors argue that detailed bio-physical models are too computational consuming. Justification is required to show that certain details are not crucial in locomotion control and thus can be safely ignored to support authors’ approach with the intermediate level of abstraction.

The model takes information from other species like zebrafish heavily. However some known experimental findings in zebrafish, like different interneuron types and their physiological behavior, is not modeled here, but applied a one type for all approach here.

The model does a finer modular neural modeling with separated trunk and tail parts. However the muscle and sensory modeling parts seem to treat them the same. Do different body parts have similar phase delay in movement and sensitivity to stimulation at one spot? Current studies do not support the uniform motor and sensory patterns. It might require extensive experimental research to get every details accurately, but it is essential to implemented model based on what we know so far, even though limited.

It is true that zebrafish can recover from spinal cord injury pretty quickly, but their locomotion pattern changes comparing to pre-injury. The authors talk about salamander SCI, but provide no results concerning it. Without RS input, will the model functions just as in real animals? Is the pattern different from pre-injury?

The article is lengthy and some parts could be simplified.

In line 280, Fig 4C is mentioned but there is no such a figure.

Extra right bracket in equation 10, making equation hard to understand if not familiar with muscle modeling.

Reviewer #2: I uploaded as an attachment.

**Have the authors made all data and (if applicable) computational code underlying the findings in their manuscript fully available?**

Reviewer #1: Yes

Reviewer #2: None

PLOS authors have the option to publish the peer review history of their article (what does this mean?). If published, this will include your full peer review and any attached files.

Reviewer #1: No

Reviewer #2: **Yes: **Asuka Takai
---

## [Decision Letter · Decision Letter 1]

10 Dec 2024

PCOMPBIOL-D-24-00688R1

Balancing central control and sensory feedback produces adaptable and robust locomotor patterns in a spiking, neuromechanical model of the salamander spinal cord

PLOS Computational Biology

Dear Dr. Pazzaglia,

Thank you for submitting your manuscript to PLOS Computational Biology. After careful consideration, we feel that it has merit but does not fully meet PLOS Computational Biology's publication criteria as it currently stands. Therefore, we invite you to submit a revised version of the manuscript that addresses the points raised during the review process.

Please submit your revised manuscript within 30 days Feb 09 2025 11:59PM. If you will need more time than this to complete your revisions, please reply to this message or contact the journal office at ploscompbiol@plos.org. Please include the following items when submitting your revised manuscript:

We look forward to receiving your revised manuscript.

Kind regards,

Thomas Serre

Section Editor

PLOS Computational Biology

Thomas Serre

Section Editor

PLOS Computational Biology

Feilim Mac Gabhann

Editor-in-Chief

PLOS Computational Biology

Jason Papin

Editor-in-Chief

PLOS Computational Biology

**Journal Requirements:**

Please ensure that the funders and grant numbers match between the Financial Disclosure field and the Funding Information tab in your submission form. Note that the funders must be provided in the same order in both places as well.

**Reviewers' comments:**

Reviewer's Responses to Questions

**Comments to the Authors:**

Reviewer #1: Thank you for addressing my concerns. I have one final question. Most of your responses, like on neuronal types, have narrowed the research down specifically to salamanders. Granted a special species. The fundamental organization, or modeling concept, has not been completely validated experimentally, and has been challenged/provided with alternative solutions, like Lindén et al. 2022. It would be helpful to discuss if your conclusions still stand should the alternative locomotion control solution is true, or the alternative is simply impossible here.

I otherwise recommend this article for publication.

Reviewer #2: I would like to thank the authors for their diligent efforts in addressing the reviewers' concerns. The addition of detailed analyses in response to Reviewer 1's Comment C1, particularly the inclusion of "Simulating Fictive Locomotion" in Supplementary File S5, appears to have addressed his/her concerns. The revisions, especially to the title, Introduction, and overall structure, have greatly improved the clarity of the authors' messages and have significantly strengthened the manuscript. This study makes a valuable contribution to understanding the neuronal basis of movement in vertebrates and will undoubtedly inspire future research.

**Have the authors made all data and (if applicable) computational code underlying the findings in their manuscript fully available?**

Reviewer #1: Yes

Reviewer #2: Yes

PLOS authors have the option to publish the peer review history of their article (what does this mean?). If published, this will include your full peer review and any attached files.

Reviewer #1: No

Reviewer #2: **Yes: **Asuka Takai

**Figure resubmission:**
---

## [Editor Report · Decision Letter 2]

26 Dec 2024

Dear MSc Pazzaglia,

We are pleased to inform you that your manuscript 'Balancing central control and sensory feedback produces adaptable and robust locomotor patterns in a spiking, neuromechanical model of the salamander spinal cord' has been provisionally accepted for publication in PLOS Computational Biology.

Best regards,

Thomas Serre

Section Editor

PLOS Computational Biology

Thomas Serre

Section Editor

PLOS Computational Biology

---

## [Editor Report · Acceptance letter]

PCOMPBIOL-D-24-00688R2

Balancing central control and sensory feedback produces adaptable and robust locomotor patterns in a spiking, neuromechanical model of the salamander spinal cord

Dear Dr Pazzaglia,

I am pleased to inform you that your manuscript has been formally accepted for publication in PLOS Computational Biology. Your manuscript is now with our production department and you will be notified of the publication date in due course.

With kind regards,

Anita Estes
